# communications
## earth & environment

# Seasonality of the Meridional Overturning Circulation in the subpolar North Atlantic

Yao Fu [1✉], M. Susan Lozier [1✉], Tiago Carrilho Biló [2], Amy S. Bower [3], Stuart A. Cunningham [4], Frédéric Cyr [5], M. Femke de Jong [6], Brad deYoung[7], Lewis Drysdale [4], Neil Fraser [4], Nora Fried [6], Heather H. Furey [3], Guoqi Han [8], Patricia Handmann [9], N. Penny Holliday [10], James Holte[2], Mark E. Inall[4,11], William E. Johns[12], Sam Jones[4], Johannes Karstensen [9], Feili Li [13], Astrid Pacini[3,15], Robert S. Pickart [3], Darren Rayner[10], Fiammetta Straneo [2] & Igor Yashayaev [14]

Understanding the variability of the Atlantic Meridional Overturning Circulation is essential for better predictions of our changing climate. Here we present an updated time series (August 2014 to June 2020) from the Overturning in the Subpolar North Atlantic Program. The 6-year time series allows us to observe the seasonality of the subpolar overturning and meridional heat and freshwater transports. The overturning peaks in late spring and reaches a minimum in early winter, with a peak-to-trough range of 9.0 Sv. The overturning seasonal timing can be explained by winter transformation and the export of dense water, modulated by a seasonally varying Ekman transport. Furthermore, over 55% of the total meridional freshwater transport variability can be explained by its seasonality, largely owing to overturning dynamics. Our results provide the first observational analysis of seasonality in the subpolar North Atlantic overturning and highlight its important contribution to the total overturning variability observed to date.

[1] School of Earth and Atmospheric Sciences, Georgia Institute of Technology, Atlanta, GA, USA. [2] Scripps Institution of Oceanography, UCSD, La Jolla, CA, USA. [3] Woods Hole Oceanographic Institution, Woods Hole, MA, USA. [4] Scottish Association for Marine Science, Oban, UK. [5] Northwest Atlantic Fisheries Centre, Fisheries and Oceans Canada, St. John's, NL, Canada. [6] NIOZ Royal Netherlands Institute for Sea Research, Texel, Netherlands. [7] Department of Physics and Physical Oceanography, Memorial University, St. John's, NL, Canada. [8] Institute of Oceans Sciences, Fisheries and Oceans Canada, Sidney, BC, Canada. [9] GEOMAR Helmholtz Centre for Ocean Research Kiel, Kiel, Germany. [10] National Oceanography Centre, Southampton, UK. [11] School of Geosciences, Edinburgh University, Edinburgh, UK. [12] Department of Ocean Sciences, University of Miami, Miami, FL, USA. [13] State Key Laboratory of Marine Environmental Science & College of Ocean and Earth Sciences, Xiamen University, Xiamen, China. [14] Bedford Institute of Oceanography, Fisheries and Oceans Canada, Dartmouth, NS, Canada. [15] Present address: Polar Science Center, Applied Physics Laboratory, University of Washington, Seattle, WA, USA. ✉email: yaofu@gatech.edu; susan.lozier@gatech.edu

Variability of the Atlantic Meridional Overturning Circulation (MOC) is broadly linked to climate variability because of its role in the global redistribution of heat and freshwater[1–3]. Despite numerous modeling studies linking MOC variability to changes in buoyancy forcing and subsequent water mass formation in the subpolar North Atlantic (SPNA)[4–6], continuous observations of MOC variability in this region were lacking prior to 2014. To fill this need, the Overturning in the Subpolar North Atlantic Program (OSNAP) deployed an observing system in the summer of 2014 to continuously measure the subpolar meridional volume transport, i.e., MOC, as well as the trans-basin meridional heat and freshwater transports (MHT and MFT, respectively)[7,8].

The OSNAP array consists of two sections——OSNAP West, which runs from the coast of Labrador to West Greenland, and OSNAP East, which runs from East Greenland to the Scottish Shelf (Fig. 1). The array includes 60 moorings, 24 on the OSNAP West section focused near the basin boundaries off Labrador and West Greenland, and 36 along the OSNAP East section with highest density off East Greenland and over the Reykjanes Ridge. The first 2 years of OSNAP observations (2014–2016) showed that overturning in the eastern subpolar gyre (OSNAP East) dominated the mean and variability of the total subpolar MOC[8], challenging the conventional view that overturning in the Labrador Sea was a major contributor to the subpolar MOC. The updated 4-year OSNAP time series (2014–2018) confirmed this result and additionally showed that the variability in any one of the SPNA deep western boundary currents (DWBC) could capture only a small portion of the total MOC variability over the observational period[9]. For OSNAP East, density changes in the interior Irminger and Iceland Basins rather than in the western boundary region appear to be important in explaining MOC variability. For OSNAP West, density anomalies in the Labrador Sea boundary currents compensate each other reducing overturning changes. These findings stand in contrast to the traditional interpretation that DWBC variability is closely linked to MOC lower limb variability[10].

One of the most striking aspects of the OSNAP MOC, MHT, and MFT time series to date has been the strong variability over the observed timescales, with a range of 15.0 Sv (1 Sv = $10^6$ m$^3$ s$^{-1}$) in MOC (with a time mean and standard error of 16.6 ± 0.7 Sv)[9], 0.2 PW in MHT (0.5 ± 0.05 PW), and 0.2 Sv in MFT (0.36 ± 0.05 Sv)[11]. Using the 71-month (~6 years) OSNAP record now in hand, the overall goal of this study is to ascertain the seasonal cycle of the subpolar MOC, MHT, and MFT in order to quantify how much of this large month-to-month variability can be attributed to seasonality.

Though MOC, MHT, and MFT seasonality across the SPNA has not previously been investigated with observations, several past studies provide useful signposts for our study. For example, an idealized two-layer model[12] was used to link overturning variability in the Labrador Sea to wintertime dense water formation. This study attributed a spring overturning peak to winter transformation and its delay to the time required for dense water in the interior to be communicated to the boundary current. Relatedly, an investigation of seasonal overturning in the Labrador Sea using Argo float data[13] found a maximum in spring and a minimum in winter. Finally, a recent study[14] revealed a fast export of newly formed Irminger Sea Intermediate Water into the OSNAP East boundary current off East Greenland. The newly formed waters reached the boundary current within 1–3 months after formation[14].

In addition to winter convection, wind forcing is also a likely cause of seasonality. Studies using data from RAPID/MOCHA (Rapid Climate Change-Meridional Overturning Circulation and Heat Flux Array–Western Boundary Time Series) have shown that the seasonal cycle of the MOC in the subtropical Atlantic is largely driven by wind forcing via direct Ekman transport changes and by isopycnal heaving at the eastern boundary[15–18]. The MOC in the subtropics peaks in fall and early winter and has a minimum in spring[16,19], with a peak-to-trough magnitude of ~4.0 Sv[20].

Over the first 4 years of OSNAP observations, the time-mean Ekman transport across the full OSNAP array is −1.5 ± 0.2 Sv ("−" indicates southward), accounting for less than 10% of the time-mean MOC (16.6 ± 0.7 Sv)[9]. In contrast to this small contribution to the mean, a recent paper[21] shows substantial seasonality in Ekman transports (−2.5 to −7.7 Sv) over the subpolar region. In light of this variability, the impact of Ekman transport on subpolar MOC seasonality will be explored here using OSNAP observations.

Even less is known about the MHT and MFT seasonality in the SPNA. OSNAP observations to date indicate that MOC variability is primarily responsible for the total MHT and MFT variability[8,11]; however, it is unclear how much of the MHT and MFT seasonal variability can be explained by MOC seasonality. That determination will be made using the new 6-year OSNAP time series.

In summary, past analyses of the OSNAP time series have focused on the determination of the mean MOC and its full spectrum of variability, the partitioning of the MOC and associated meridional heat and freshwater transports between the two sections, and the relationship between MOC variability and western boundary current variability. With this new 6-year time

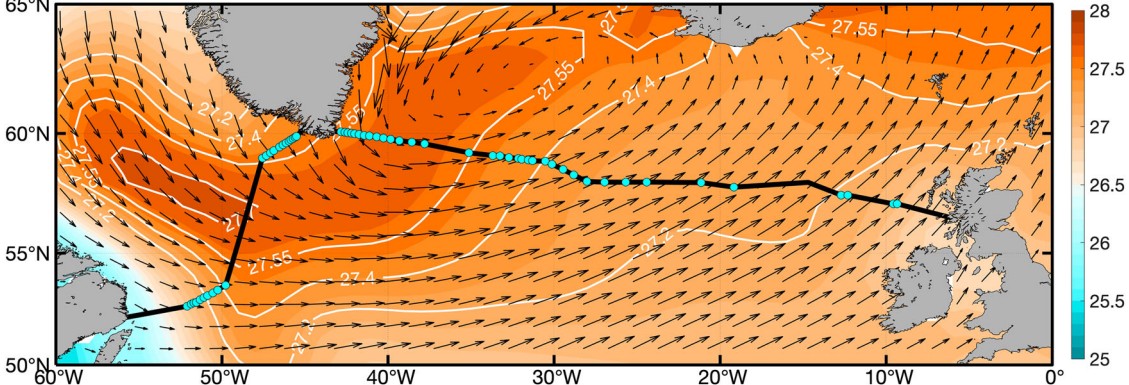

**Fig. 1 OSNAP Section.** Winter (January–March) climatological wind stress from ERA5 (2014–2020) over the subpolar North Atlantic (vectors, in N m$^{-2}$) superimposed on climatological winter sea surface density from EN4 (shading with contours, in kg m$^{-3}$, for 2014–2020). OSNAP West and OSNAP East sections are indicated by black lines with OSNAP mooring positions denoted with cyan dots.

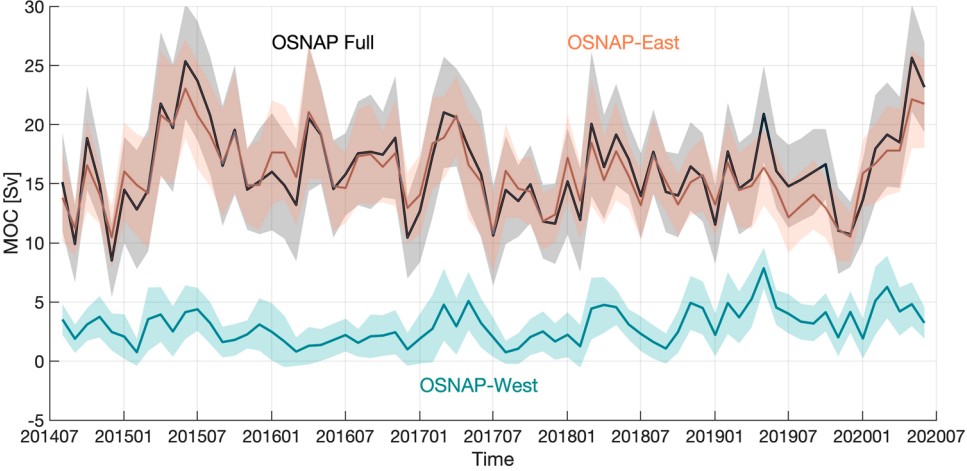

**Fig. 2 Meridional Overturning Circulation (MOC) time series.** Monthly MOC time series across the full OSNAP array (black), OSNAP East (orange), and OSNAP West (cyan). Shading indicates monthly uncertainty estimated from a Monte Carlo analysis[8].

series, we focus on determining the MOC, MHT, and MFT seasonality and investigate their relationship to the wintertime transformation of water masses and seasonal wind-forced Ekman transport.

## Results

The retrieval of mooring data in 2020 extended the OSNAP observation time series to 71 months, covering the period from August 2014 to June 2020 (Fig. 2). In the following, we first present the 6-year MOC time series, from which we determine MOC seasonality and explore how winter transformation and wind-driven Ekman transport impacts that variability. The 6-year MHT and MFT time series and their corresponding seasonality are then investigated.

**MOC time series.** Over the 6 years of OSNAP observations, the MOC across the full OSNAP array exhibits strong monthly to interannual variability, with a range of 10–25 Sv and a standard deviation of 3.7 Sv. The mean MOC (±standard error) is 16.7 ± 0.6 Sv. The standard error is estimated based on the effective number of degrees of freedom, taking into account the autocorrelation of the time series (see "Methods"). As reported previously[8,9], the mean strength and variability of the MOC are dominated by overturning across OSNAP East (16.3 ± 0.6 Sv, with a standard deviation of 2.9 Sv) rather than across OSNAP West (3.0 ± 0.5 Sv, with a standard deviation of 1.5 Sv). Following the publication of the first OSNAP time series, the observed partitioning of the total MOC between the western and eastern subpolar basins has been reproduced by a few newer-generation climate models[4,22,23].

The two additional years in this time series (June 2018–June 2020) reveal that the overturning at OSNAP East was relatively weak throughout 2019 and then rose sharply in the first half of 2020. Correspondingly, the MOC across the full array reached a maximum in May 2020 (26.4 ± 4.6 Sv) with comparable strength to that in 2015 (25.6 ± 4.8 Sv). The overturning at OSNAP West was relatively strong in both 2019 and 2020 compared to that in the first 4 years (Supplementary Table 1). The mean OSNAP West MOC for the last 2 years (August 2018–June 2020) is 3.9 ± 0.9 Sv, substantially stronger than the first 2-year mean (2.5 ± 0.6 Sv). The strongest monthly OSNAP West overturning, in May 2019, reaches 8.1 ± 1.7 Sv and contributes strongly to the MOC peak across the full OSNAP array in that month.

**MOC seasonal cycle.** A seasonal cycle of the SPNA MOC is extracted by calculating the climatological mean MOC for each month using the 6-year OSNAP time series (Fig. 3a, see "Methods"). Across the full OSNAP section, the seasonal MOC maximum in spring (21.1 ± 1.2 Sv) is significantly larger than the minimum in winter (12.3 ± 1.2) with a peak-to-trough change of 8.8 Sv (Supplementary Table 2, see "Methods" for the standard error estimation). Overall, seasonal variability explains about 40% of the total variability of the monthly MOC time series across the full OSNAP array over the 6-year observational period.

The MOC seasonal phasing for OSNAP East and OSNAP West is similar to that across the full array (Fig. 3a). Consistent with the variability of the entire (6-year) time series, seasonal variability of the full MOC is dominated by OSNAP East, with OSNAP West playing a lesser role. The range of the seasonal cycle is 6.2 Sv for OSNAP East and 2.6 Sv for OSNAP West, both with a maximum in May, but with the minimum at OSNAP West occurring in September rather than in December as it does for both OSNAP East and the full basin MOC. Overall, the seasonal cycle explains about 36% of the total variability of the monthly overturning at OSNAP East and about 25% at OSNAP West over the 6-year observational period. Note that neither the explained variability by the MOC seasonality at OSNAP East or West exceeds that across the full array. This can be explained by an enhanced MOC seasonality due to a combination of OSNAP East and OSNAP West MOC with their seasonality expressed over different periods. For instance, the OSNAP East MOC exhibits spring peaks over the observational period except in 2019, while the OSNAP West MOC only shows a distinct seasonal cycle after 2017 (Fig. 2). Although the MOC across the full array is not a linear sum of the MOC across OSNAP East and OSNAP West, the combination still results in an enhanced MOC seasonality across the full array.

**Wintertime transformation.** North Atlantic Deep Water (NADW) is the primary source of the MOC lower limb. Therefore, the formation and southward transport of NADW in the SPNA has been linked to the intensity and variability of the MOC[24]. To investigate the linkage between transformation and the MOC on seasonal timescales, we evaluate the seasonal cycle of water mass transformation (see "Methods") across the 27.54 kg m$^{-3}$ and 27.69 kg m$^{-3}$ potential density ($\sigma_\theta$) isopycnals in the eastern subpolar gyre (i.e., Irminger and Iceland Basins) and western subpolar gyre (Labrador Basin), respectively (Fig. 3b). Positive transformation indicates water is converted

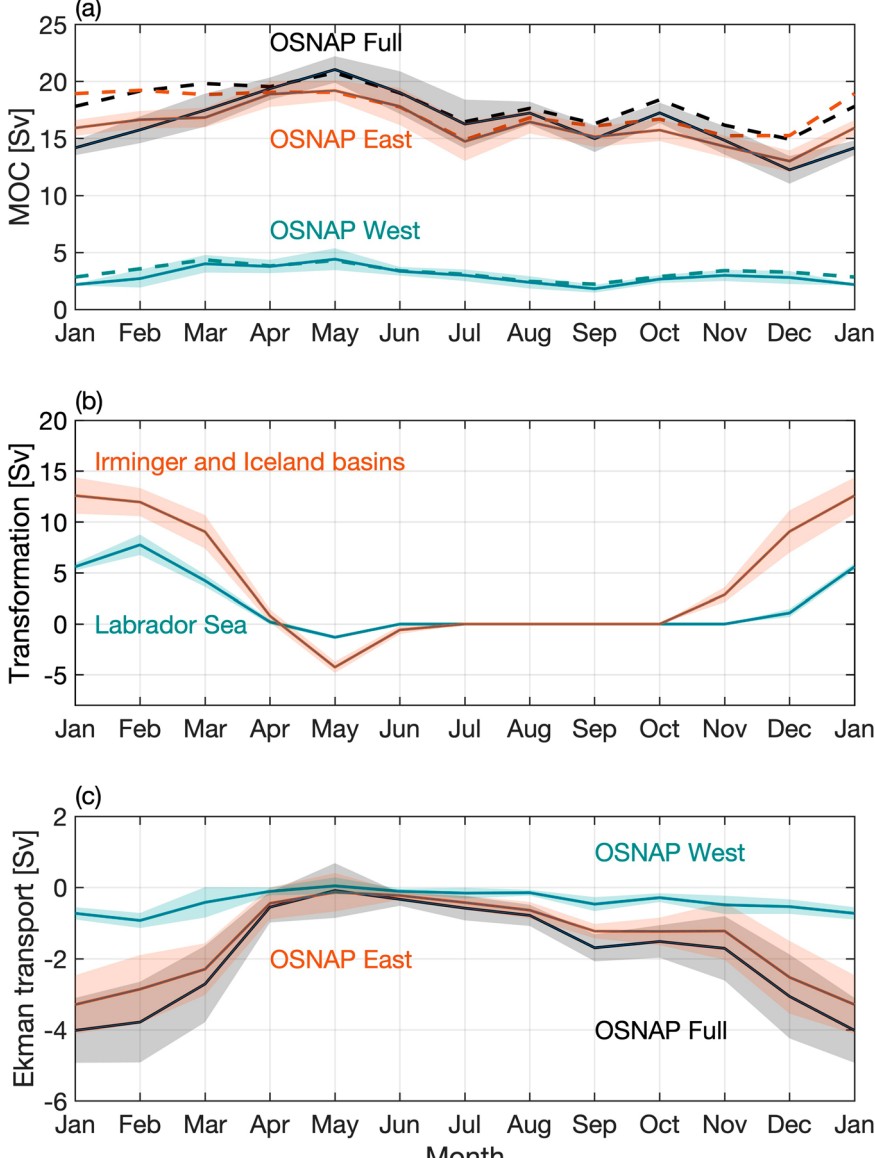

**Fig. 3 Seasonal cycle of the Meridional Overturning Circulation, transformation, and Ekman transport.** Monthly climatology of **a** the MOC across the full OSNAP array (solid black), OSNAP East (solid orange), and OSNAP West (solid cyan), **b** transformation in the OSNAP East (orange) and OSNAP West (cyan) regions, and **c** Ekman transport across the full OSNAP array (black), OSNAP East (orange) and OSNAP West (cyan). The OSNAP West region is defined as the area between the 2000-m isobath of the Labrador Sea north of the OSNAP West line. The OSNAP East region is defined as the area between the OSNAP East section and the Greenland-Scotland-Ridge. Shading in the plots indicates the standard error of the monthly climatology over the 6-year OSNAP observation period. Ekman transport across the OSNAP array is calculated by projecting the total wind onto the OSNAP West and East sections. Dashed lines in **a** indicate the MOC seasonal cycle with both the Ekman transport and Ekman transport return flow removed (see "Methods").

from lower density to higher density due to air-sea heat and freshwater fluxes. The two isopycnals are the time-mean potential density surfaces of maximum overturning ($\sigma_{MOC}$) at OSNAP East and OSNAP West, respectively. Note that the MOC is calculated as the maximum of the overturning stream function, which occurs at different $\sigma_{MOC}$ at different time steps. When the transformation is calculated at time-varying $\sigma_{MOC}$ for each month, the derived sea-sonality of transformation for both OSNAP East and OSNAP West region is qualitatively similar to that calculated at the time-mean $\sigma_{MOC}$. Transformation in both the eastern and western subpolar gyre shows a distinct peak in winter from December to March and remains near zero during the other months of the year, except in spring when re-stratification of the water column through surface warming reduces surface water density leading to negative trans-formation (Fig. 3b).

Overturning across both OSNAP East and OSNAP West peaks in May which is 3–5 months after the maximum transformation, and then slowly declines over the rest of the year (Fig. 3a, solid lines). The OSNAP East overturning seasonality can be partly attributed to the fast export of Irminger Sea Intermediate Water (27.65 to 27.77 kg m$^{-3}$) by the East Greenland Current[14]. As previously reported, this water mass leaves the OSNAP East section within 1–3 months after formation off East Greenland[14], as evidenced by enhanced southward East Greenland Current transport from December to May[9,14]. The spring overturning peak at OSNAP West matches the arrival of southward propagating density anomalies along the Labrador Sea western boundary at the OSNAP West array (Supplementary Fig. 1). This timing is consistent with that predicted by Straneo et al.[12] and highlights the expected importance of buoyancy forcing on

seasonal timescales. In addition, the observed oxygen concentration in the LSW layer at OSNAP West also peaks in spring[25], consistent with the observed overturning maximum. The results above collectively indicate that wintertime formation and the export of dense water leads to increased overturning in winter at both OSNAP East and West.

**Ekman transport.** We now assess the impact of Ekman transport (Eq. 3c) on MOC seasonal variability (see "Methods"). The 6-year mean Ekman transport (±standard error) across the full OSNAP array is southward, at $-1.7 \pm 0.3$ Sv, with the eastern subpolar gyre (OSNAP East) contributing $-1.4 \pm 0.2$ Sv to the total, and the western subpolar gyre (OSNAP West) contributing $-0.4 \pm 0.1$ Sv. These mean values are overshadowed by the large range of Ekman transports over its climatological seasonal cycle (Fig. 3c). Given that the westerly winds over the SPNA are strongest in winter and weakest in summer, we find the expected seasonality in Ekman transport. The monthly climatology of Ekman transport across the full array, always southward, has its maximum magnitude ($-3.0$ to $-4.1$ Sv) in winter (December–February) and its minimum ($-0.1$ to $-0.3$ Sv) in spring through summer (April–August) (Supplementary Table 3).

The Ekman transport has a different impact on the two sections owing to their different orientation relative to the westerly winds: while the OSNAP East line is nominally zonal, OSNAP West is nearly orthogonal to the winds (Fig. 1). In addition, the OSNAP East section covers a much longer distance (~1970 km) than the OSNAP West section does (~890 km). As such, the impact of Ekman transport is pronounced for the OSNAP East overturning yet of little consequence for the OSNAP West overturning. In particular, the seasonal cycle of Ekman transport enhances the overturning seasonal cycle at OSNAP East (and consequently across the full OSNAP array) by weakening the MOC in winter, when it opposes the generally northward flow of the upper ocean across the OSNAP line. In spring (May), the Ekman transport is nearly zero at both sections and has little impact on the MOC. As a result, the Ekman transport explains the timing of the MOC peak across the full array (Fig. 3a, dashed lines, Supplementary Table 2). Finally, we note that the seasonal impact of Ekman dynamics is insensitive to changes in the latitudinal position of OSNAP East of ±1° (see "Methods").

**Summary of MOC seasonal variability.** Overall, wintertime dense water formation and Ekman transport seasonality are two major contributors to MOC seasonal variability across the full OSNAP array. The wintertime dense water formation and export are responsible for the MOC peak in spring at both OSNAP East and OSNAP West, while the maximum southward Ekman transport in winter contributes substantially to the MOC minimum, which primarily applies to OSNAP East overturning, and thus to the full MOC.

An advantage of assessing the seasonal cycle is that its removal from the monthly MOC time series reveals the MOC interannual variability (Supplementary Fig. 2). At OSNAP East, a strong MOC peak in 2015 is particularly evident, as are the weak MOC across OSNAP West in 2016 and the strong MOCs in 2019 and 2020 also across OSNAP West. An investigation of the mechanisms driving the SPNA MOC variability based on this OSNAP time series is underway.

**Meridional heat and freshwater transports.** Over the 6-year OSNAP observational period, the MHT has a mean (±standard error) of $0.50 \pm 0.01$ PW. The eastern and western subpolar gyres contribute $0.42 \pm 0.01$ PW and $0.08 \pm 0.01$ PW, respectively, to this total. The heat transport across the section extending from

the Reykjanes Ridge across the Iceland Basin to the Scottish continental shelf dominates the total MHT (Fig. 4a, red), which is attributable to the northward transport of warm subtropical waters by the North Atlantic Current (NAC). The MHT exhibits strong monthly to interannual variability, which is clearly dominated by variability at OSNAP East (Fig. 5a). A decomposition of MHT into an overturning component and a gyre component[8] indicates that the overturning circulation accounts for 73% of the total subpolar MHT variability and the remaining part is attributed to gyre circulation. At OSNAP East, overturning dynamics explain ~90% of the MHT variability. At OSNAP West, overturning and gyre circulation dynamics each explain about 40% of the MHT variability.

The 6-year mean MFT (±standard error) across the full OSNAP array is $-0.36 \pm 0.01$ Sv. In contrast to the MOC and MHT, MFT across OSNAP West ($-0.17 \pm 0.01$ Sv) and OSNAP East ($-0.18 \pm 0.01$ Sv) contributes similarly to the total MFT with respect to both the mean and variability (Fig. 5b). This highlights the importance of the Labrador Basin in terms of ocean freshwater transport, especially considering that the Labrador Sea has a smaller basin size and weaker overturning compared to the eastern subpolar gyre[8]. The freshwater transport at OSNAP West is primarily concentrated in the Labrador Sea western boundary region (Fig. 4a, blue; see "Methods"), where the southward export of very fresh coastal waters (Fig. 4b) plays an important role[11]. At OSNAP East, both the western boundary and NAC regions contribute to the southward freshwater transport, with the former related to the southward transport of fresh coastal waters and the latter due to the northward transport of salty waters by the NAC, which is equivalent to a southward freshwater transport. The overturning component is responsible for ~53% of the subpolar MFT variability across the full OSNAP array, 66% across OSNAP East, and 87% across OSNAP West.

**Seasonal variability of MHT.** Compared to the MOC seasonality, MHT seasonality is relatively weak (Fig. 6). Across the full OSNAP array, MHT varies from 0.45 to 0.55 PW. The seasonal maximum to minimum range of 0.10 PW is only 17% of the mean MHT, while the seasonal range of the MOC (~9.0 Sv) is ~50% of the mean. Only 21% of the total MHT variability across the full OSNAP array is explained by the seasonality, compared to (as mentioned above) 40% of the total MOC. MHT across OSNAP East ranges from 0.36 to 0.46 PW and across OSNAP West from 0.06 to 0.11 PW.

Seasonal MHT variability across the full OSNAP array mirrors that across OSNAP East. Both time series are marked by two maxima—one in spring and the other in autumn, and a minimum in summer (Fig. 6). MHT at OSNAP West, with a maximum in winter and minimum in spring, plays a minor role in seasonal MHT variability across the full array. Ekman transport at OSNAP East, strongest in winter, carries relatively warm surface water southward, thus reducing the total northward MHT. As a result, the Ekman transport contribution to MHT greatly reduces the MHT seasonal variability at OSNAP East and across the full array, while it has little impact at OSNAP West (Fig. 6, dashed lines). The average difference of MHT from December to March due to Ekman transport contribution is about 0.08 PW for the full array and for OSNAP East.

A decomposition of MHT variability into that due to overturning variability and that due to gyre variability helps explain the extrema noted above for OSNAP East and OSNAP West. The overturning contribution (inclusive of the Ekman contribution) is primarily responsible for MHT seasonal variability at OSNAP East (Fig. 7a), with a maximum northward heat transport in spring, minimum in winter, and weaker

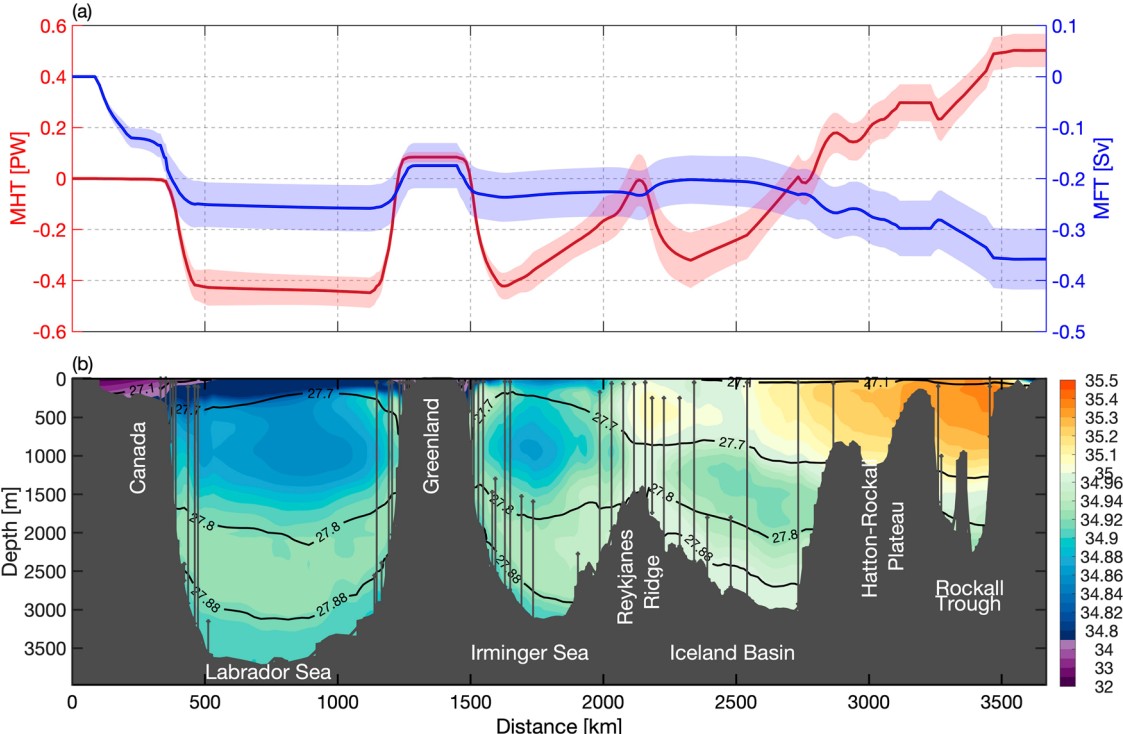

**Fig. 4 Salinity and meridional heat and freshwater transports across the OSNAP section. a** Cumulative MHT (red) and MFT (blue) integrated from west to east. Northward transport is defined as positive. Shading indicates the corresponding standard deviation over the 6-year observational period. **b** The 6-year mean salinity section (colored shading) with moorings marked by the vertical black lines. The horizontal black lines represent the isopycnals of 27.10, 27.70, 27.80, and 27.88 kg m$^{-3}$.

secondary maximum in October. The maximum of OSNAP East MHT in spring is more muted than that of the OSNAP East MOC because the waters moving northward in the upper limb of the MOC are at their coldest point of the year in spring (March and April), offsetting the impact of increased overturning strength in spring. Similarly, in terms of the OSNAP East total MHT, the warmer upper ocean temperatures in late summer to early autumn (July to October) partially offset the reduction in the MOC at that time relative to spring. The gyre contribution is relatively weak from January through September but strengthens in October and November due to an enhanced temperature difference between the western and eastern parts of the eastern subpolar basins in those months as well as enhanced gyre circulation in winter. The enhanced gyre heat transport also contributes to the second maximum of MHT across the full array in autumn. These two effects: the annual variation in upper limb average temperatures, and the gyre contribution to the MHT, largely explain why the MHT seasonal cycle is so much weaker than the MOC seasonal cycle.

At OSNAP West, both the overturning and gyre components contribute strongly to the mean and variability of MHT on seasonal timescales (Fig. 7b). Unlike OSNAP West MOC seasonal cycle, the overturning MHT component does not show a peak in spring because the waters in the upper limb are coldest in early spring (March–April), reducing the northward heat transport by the MOC. The gyre component is weak in late winter and early spring (February–May) and strong in summer (July–September). This seasonal variation is largely attributed to the temperature contrast between the western and eastern half of the Labrador Basin, which is strongest in summer and weakest in late winter and spring.

**Seasonal variability of MFT**. Over 55% of the total MFT variability across the full OSNAP array can be explained by the

seasonality over the observational period. The proportion even reaches 66% at OSNAP West, while it is about 37% at OSNAP East.

MFT across the full OSNAP array is strongest in winter and weakest in spring through summer (Fig. 8). Note that MFT is southward all year round. Here we refer to the strongest southward MFT as maximum and the weakest southward MFT as minimum. Distinct from the MOC and MHT, freshwater transport across both OSNAP East and OSNAP West contributes strongly to the MFT seasonal variability across the full OSNAP array. At OSNAP East, MFT is strongest from October to February, followed by a gradual decline until the summer minimum (with a seasonal range from −0.22 to −0.13 Sv). At OSNAP West, the MFT seasonal time series has a pattern similar to that for MFT across the full array, with a maximum in winter and a minimum in spring. Its seasonal range is from −0.24 to −0.11 Sv. The seasonal Ekman transport component has only a marginal impact on the MFT seasonality across the full array, OSNAP East and OSNAP West (Fig. 8, dashed lines). This is attributed to two factors: (1) the MFT seasonal cycle across the full array is more strongly expressed at OSNAP West, where the Ekman transport is weak throughout the year. (2) For OSNAP East, the section-mean salinity within the Ekman depth (about 25 m averaged over the section) over the OSNAP observation period is 35.01, close to the full-array mean salinity of 34.92 (see "Methods"). This minimizes the contribution of Ekman transport to MFT at OSNAP East.

The decomposed MFT seasonal cycle at OSNAP East reveals that the overturning component is primarily responsible for the total MFT seasonality (Fig. 9a). The strong overturning component from January to April can be attributed to (1) a strengthened northward transport of the upper limb, which carries the warm salty waters of the NAC and (2) an enhanced southward transport of relatively fresh water mainly along the

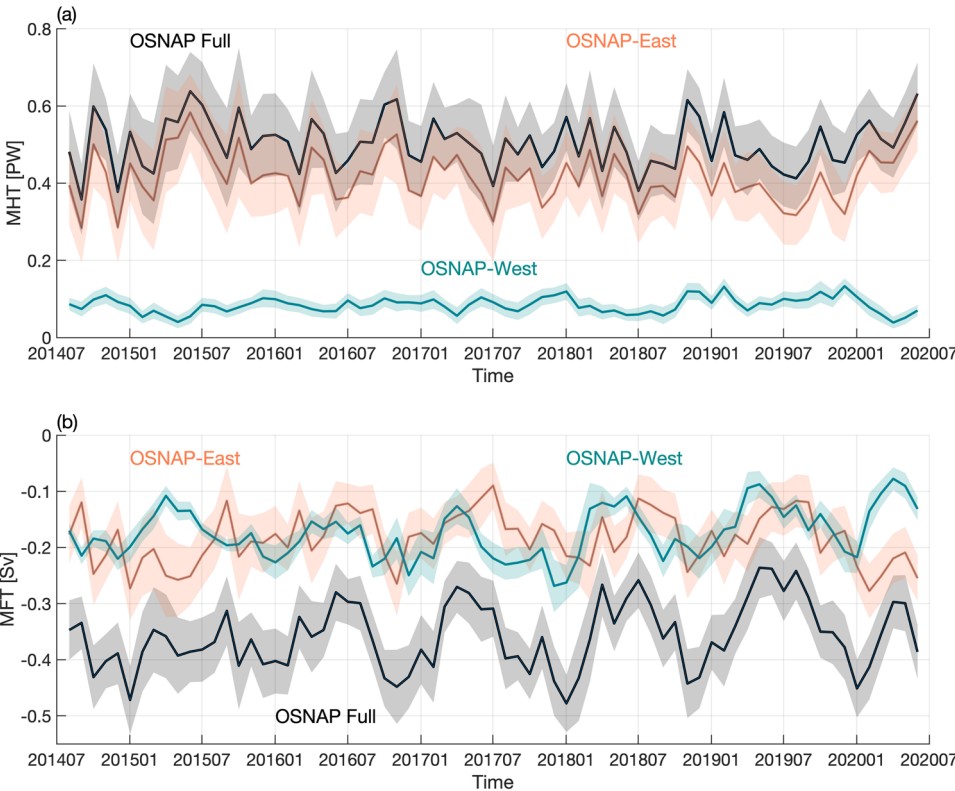

**Fig. 5 Meridional heat and freshwater transport time series.** Monthly MHT (**a**) and MFT (**b**) across the full OSNAP array (black), OSNAP East (orange), and OSNAP West (cyan). Shading indicates the monthly uncertainty estimated using a Monte Carlo analysis[8].

western boundary of the Irminger Sea. In summer, freshening of the upper ocean due to land/sea ice melt contributes to the stratification of the water column, which results in a weakening of the local overturning. As for the gyre MFT component, it has an enhanced contribution in October and November, when the salinity contrast between the Iceland-Rockall Trough Basins and the Irminger Basin is at its peak. Gentle summer winds ease the gyre circulation and hence the gyre contribution to the MFT. Together, the overturning and gyre components both contribute minimally to MFT in summer.

At OSNAP West, the overturning component dominates the total MFT seasonal variability with a maximum in winter and a minimum in spring (Fig. 9b). An analysis of the MFT overturning component at OSNAP West indicates that its seasonality is largely expressed in density classes lighter than 27.1 kg m$^{-3}$, which indicates that the fresh waters along the Labrador Sea shelf (Fig. 4b) play a major role in setting this seasonality. The gyre component plays a secondary role, with maximum southward freshwater transport in late summer and minimum in late winter and early spring. The strongest gyre contribution in summer is due to the export of very fresh water along the Canadian shelf and slope at the western boundary of the Labrador Sea, likely resulting from land/sea ice melt.

## Summary

The extended 6-year OSNAP observations reveal robust seasonal variability of the MOC, MHT, and MFT in the SPNA, but particularly so for MOC and MFT, for which seasonal variability accounts for a large fraction of the total variability. The MOC across both OSNAP East and OSNAP West peaks in spring and has a minimum in winter. This seasonality is largely driven by the wintertime formation and export of dense water and by the Ekman transport seasonal cycle. By reducing the northward transport in the upper layer primarily in winter, Ekman dynamics delay the appearance of the MOC peak, thus enhancing the MOC seasonal cycle across OSNAP East and the full array. The seasonal cycle of the MOC across the full OSNAP array has a magnitude of ~8.8 Sv and explains ~40% of the total MOC variability during the 6-year observational period. In comparison, the MOC seasonal cycle in the subtropics has a range of ~4.0 Sv, with a maximum in autumn and a minimum in spring.

During the 6-year OSNAP observational period (2014–2020), the winter North Atlantic Oscillation (NAO) was persistently in a positive phase with strong convection in the Labrador and Irminger Seas[26,27]. Because a positive winter NAO is associated with strong westerly winds[28], we expect the seasonality of the Ekman transport to be amplified under these conditions. Thus, our observations to date may cover a period during which the Ekman seasonality has a maximum impact on the MOC. With the shifting phase of the NAO and long-term climate change, the atmospheric conditions that drive the Ekman transport and winter transformation will likely also change. Therefore, the magnitude of the seasonal variability may change as more years are added to this record. However, given the inherent seasonality of atmospheric forcing, i.e., the westerly winds and surface buoyancy forcing, we do not expect the phasing of the seasonal cycles presented here to fundamentally change.

In addition, modeling studies[29–33] have indicated that on interannual to decadal timescales, positive and negative NAO phases are linked to stronger and weaker MOCs, respectively. We expect that Ekman dynamics and transformation during the different NAO phases will impact MOC variability on those same timescales. Moreover, wind variability can impact MOC variability through the generation of wind stress anomalies in the central and eastern SPNA that spawn baroclinic Rossby waves. These waves disturb isopycnals in their westward propagation[34], creating a baroclinic response of the velocity field on interannual

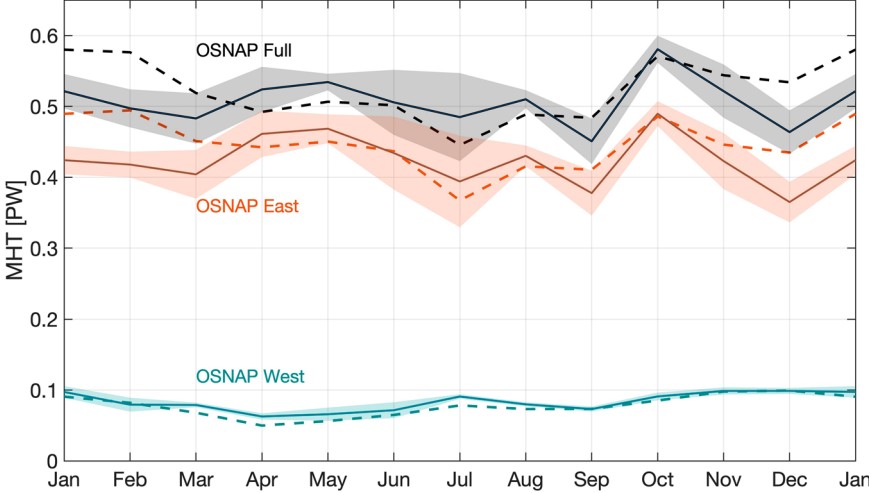

**Fig. 6 Seasonal cycle of meridional heat transport.** Monthly climatology of MHT across the full OSNAP array (black), OSNAP East (orange), and OSNAP West (cyan). Shading shows the standard error of the monthly MHT climatology based on the 6-year OSNAP observation. Dashed lines indicate the MHT seasonal cycle after the removal of the Ekman transport contribution.

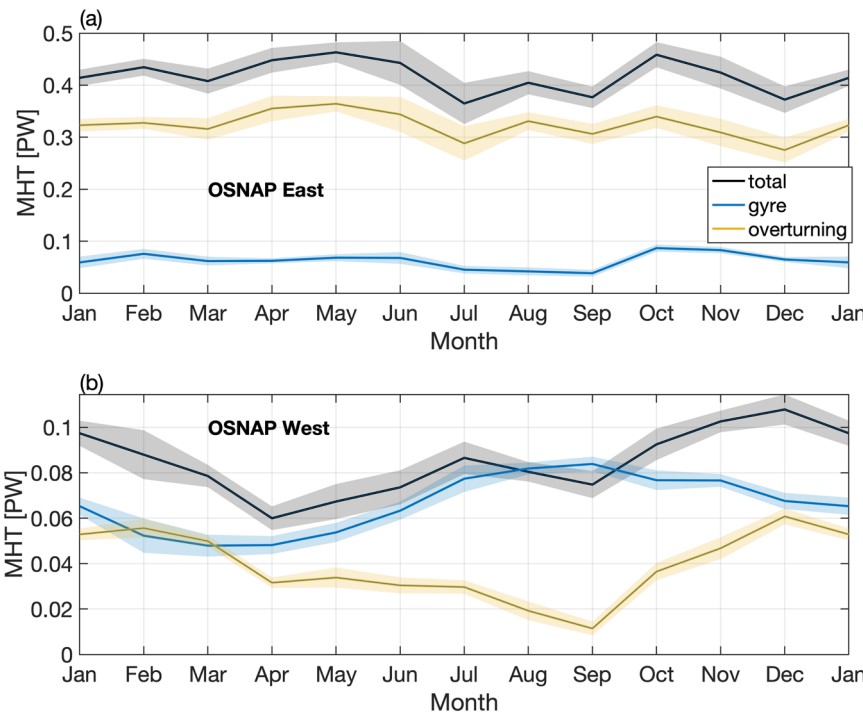

**Fig. 7 Decomposition of seasonal meridional heat transport.** Decomposition of seasonal MHT at OSNAP East (**a**) and OSNAP West (**b**) into an overturning component (yellow) and a gyre component (blue). The total MHT is shown in black. Shading indicates the standard error of the monthly climatology.

timescales. Our next 2-year observational period will be telling since the NAO index was negative for the winter of 2021, offering us an opportunity to examine interannual variability of the subpolar MOC and its link to NAO-related wind and buoyancy forcing changes.

The MHT across the OSNAP array exhibits weak seasonal variability due in large part to strong southward Ekman transport in winter, which offsets the northward flow of warm water in the MOC upper limb in the Iceland Basin. This moderation of the MHT seasonal variability by the Ekman transport likely has implications for regional weather and ecosystems in the high-latitude North Atlantic region. Any moderating role for Ekman transport in the determination of MHT variability on longer

timescales (i.e., interannual to multidecadal) is yet to be determined but is clearly of interest.

MFT's strong seasonal variability is primarily driven by the export of fresh and light coastal waters along the western boundary in the western subpolar gyre and is further amplified in winter and spring due to the enhanced transport of salty waters carried by the NAC. The dominance of the seasonal signal in the total MFT variability is largely expressed at OSNAP West. Though the Labrador Sea only weakly contributes to the MOC and MHT, this strong contribution to MFT highlights its potentially important role in future climate change. Studies have indicated that melt waters transported along the eastern boundary of the Labrador Sea contribute a

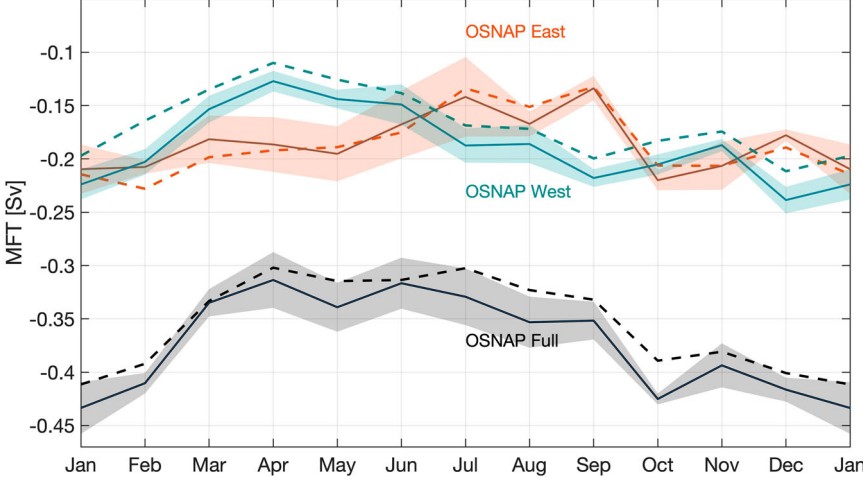

**Fig. 8 Seasonal cycle of meridional freshwater transport.** Monthly climatology of MFT across the full OSNAP array (black), OSNAP East (orange), and OSNAP West (cyan). Shading indicates the standard error of the monthly MFT climatology based on the 6-year OSNAP observations. Dashed lines indicate MFT seasonal cycle after the removal of the Ekman transport contribution.

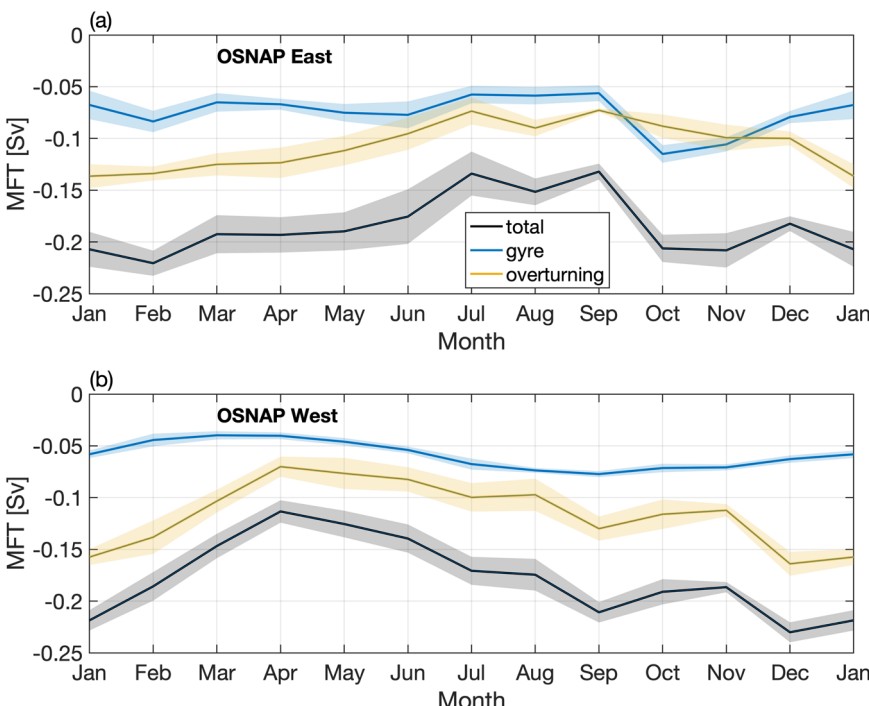

**Fig. 9 Decomposition of seasonal meridional freshwater transport.** Decomposition of seasonal MFT at OSNAP East (**a**) and OSNAP West (**b**) into an overturning component (yellow) and a gyre component (blue). The total MFT is shown in black. Shading indicates the standard error of the monthly climatology.

large portion of freshwater to the Labrador interior basin[35,36]. As such, the expected introduction of large amounts of freshwater from glaciers melting into the Labrador Basin in the years ahead poses an increased threat of a shutdown of deep convection[37]. Continuation of the OSNAP observations will help us monitor and understand the relationship between deep convection and glacial melt, as well as unravel the complex relationship between Arctic sea ice and the MOC[38–41]. Finally, though the Labrador Basin has been highlighted here because of its outsized role in MFT, it is actually the eastern subpolar basin that is currently experiencing a record-breaking freshening event[42]. The freshening signal has arrived in the Irminger Basin[43], where it is expected to weaken deep convection in the coming winters.

In summary, following earlier published studies of the OSNAP time series[8,9], this current extension enhances our understanding of the SPNA overturning circulation with its focus on seasonality. The MOC, MHT, and MFT seasonality established here provides an observational basis for further studies on overturning variability on other timescales and for model ground-truthing. This ground-truthing is needed because many state-of-the-art climate models still show discrepancies in simulating the subpolar MOC due to biases in the frequency, strength, and location of convection in the subpolar region[44,45]. The OSNAP mooring turn-around in the summer of 2022 will extend the time series to 8 years, at which point we plan to investigate the mechanisms responsible for subpolar overturning variability on interannual to sub-decadal timescales. With a likely weakening of the MOC in

the coming decades[46], monitoring MOC variability and its connection to heat, freshwater, oxygen and carbon exports are vital for deepening our understanding of global climate variability.

## Methods

**MOC, MHT, and MFT calculations.** Following previous studies[8,9], the MOC is defined as the maximum of the overturning stream function, $\Psi$, in potential density ($\sigma_\theta$) space as:

$$\text{MOC}(t) = \max[\Psi(\sigma, t)] = \max\left[\int_{\sigma_{\min}}^{\sigma_{\max}} \int_{x_w}^{x_e} v(x, \sigma, t)dxd\sigma\right], \quad (1)$$

where $v$ is the cross-sectional volume transport per unit length per unit density along the OSNAP section (positive poleward), which is integrated from west ($x_w$) to east ($x_e$) and from the smallest density surface ($\sigma_{\min}$) to the largest ($\sigma_{\max}$) throughout the water column. The velocity field along the OSNAP section is derived using a combination of direct velocity measurements and geostrophic calculation based on the thermal wind relation[47]. The density at which the overturning stream function reaches the maximum is $\sigma_{MOC}$. The MOC upper (lower) limb transport is the transport above (below) $\sigma_{MOC}$ in the overturning stream function. The 2014–2020 mean $\sigma_{MOC}$ is 27.63 kg m$^{-3}$ across the full OSNAP section, 27.54 kg m$^{-3}$ across OSNAP East, and 27.69 kg m$^{-3}$ across OSNAP West.

MHT is defined as:

$$\text{MHT}(t) = \rho C_p \int_{\sigma_{\min}}^{\sigma_{\max}} \int_{x_w}^{x_e} v(x, \sigma, t)\,\theta(x, \sigma, t)dxd\sigma, \quad (2)$$

where $\rho$ is potential density, $C_p$ is the specific heat of seawater, and $\theta$ is potential temperature.

MFT is defined as:

$$\text{MFT}(t) = -\int_{\sigma_{\min}}^{\sigma_{\max}} \int_{x_w}^{x_e} v(x, \sigma, t)\frac{S(x, \sigma, t) - \bar{S}}{\bar{S}} dxd\sigma, \quad (3)$$

where $S$ is salinity and $\bar{S}$ is the area-weighted section-mean salinity (34.92 across the full OSNAP section). Note that MFT is defined following Lozier et al.[8] and should be regarded as an equivalent oceanic freshwater transport due to meridional salinity transport.

**Ekman transport calculation.** The cross-sectional Ekman transport for the OSNAP section, $M_{Ek}^y$, is calculated from ERA5 surface wind stress as:

$$M_{Ek}^y = -\frac{1}{\rho_0}\frac{\tau_x}{f}, \quad (4)$$

where $\rho_0$ is the density of seawater with a constant value of 1025 kg m$^{-3}$, $\tau_x$ is the wind stress projected along the OSNAP section, and $f$ is the Coriolis parameter. The Ekman transport is applied to the OSNAP section where there is no direct velocity measurement near the surface[8] and within the theoretical Ekman layer[48]. The section-mean Ekman layer depth is about ~25 m.

The zero net volume transport constraint across the full OSNAP section requires the surface-layer Ekman transport to be balanced by an instantaneous barotropic return flow that is evenly distributed across the full section. In the calculation of the MOC, MHT, and MFT without Ekman transport components, we removed the Ekman transport in the Ekman layer and the Ekman return flow from the derived velocity field at each time step.

Ekman transport across OSNAP East dominates the mean and variability of the total Ekman transport across the full array. This dominance can be attributed to the fact that the orientation of the OSNAP East section is almost parallel to the westerly winds, which generates large cross-sectional Ekman transport. To test the sensitivity of the Ekman transport strength to the latitudinal location of the OSNAP East, we randomly shifted the entire OSNAP East section within 1° latitude to the north or south of its current position. These runs yielded mean Ekman transports within 0.1 Sv of the mean Ekman transport (−1.7 ± 0.3 Sv) across the current line, leading us to conclude that sensitivity to local latitudinal variability is weak.

**Water mass transformation rate.** Following previous studies[24,49], the rate of water mass transformation, $F$, is calculated by integrating surface density fluxes induced by air-sea buoyancy fluxes over the outcropping area of a given isopycnal as:

$$F(\sigma^*) = \frac{1}{\Delta\sigma}\iint\left[-\frac{\alpha}{C_p}Q + \beta\frac{S}{1-S}(E-P)\right]\prod \sigma dxdy, \quad (5)$$

where

$$\prod \sigma = \begin{cases} 1 & \text{for } [\sigma - \sigma^*] \le \frac{\Delta\sigma}{2} \\ 0 & \text{elsewhere.} \end{cases}$$

In Eq. (5), $\alpha$ and $\beta$ are the thermal expansion and haline contraction coefficients, respectively. $Q$ is the net heat flux into the ocean, and $S$ is the surface salinity. $E$ and $P$ are the evaporation and precipitation rates, respectively. $\Delta\sigma = 0.2$ kg m$^{-3}$ is the

density bin size that the outcropping density ($\sigma^*$) represents. Positive transformation indicates the conversion of water lighter than the specified isopycnal to water denser than the isopycnal. The Labrador Basin is defined as the area enclosed by the 2000-m isobath of the Labrador Sea and the OSNAP West line. Using the 2000-m isobath excludes the shallow continental shelf region along the Labrador coast, where water is much lighter than the mean $\sigma_{MOC}$ of 27.69 kg m$^{-3}$ in the Labrador Sea. The continental slope on both sides of Greenland is very steep. Using the coastline or 2000-m isobath has negligible impact on the outcropping area. The eastern subpolar gyre is defined as the area between the OSNAP East section and the Greenland-Scotland-Ridge[24].

The National Centers for Environmental Prediction Climate Forecast System v2 (NCEP/CFSv2) heat and freshwater fluxes[50] and Roemmich-Gilson Argo Climatology monthly temperature and salinity[51] from January 2014 to December 2020 are used to calculate the transformation. Using a different combination of datasets, for example, NCEP/CFSv2 heat and freshwater fluxes with surface temperature and salinity derived from the Met Office EN4[52], results in a qualitatively similar transformation seasonal cycle, with the same phase yet a slightly different winter transformation. Specifically, the December–March mean transformation (±standard deviation) for the eastern subpolar gyre is 10.7 ± 1.9 Sv and 11.8 ± 1.8 Sv for the Argo-based and EN4-based calculations, respectively. The values for the western subpolar gyre are 4.6 ± 2.8 Sv and 4.3 ± 1.9 Sv for the Argo-based and EN4-based calculations, respectively. Given the similarity of these estimates, we use the combination of NCEP/CFSv2 and Argo to demonstrate the seasonal cycle of water mass transformation.

**Monthly climatology calculation.** The climatological mean MOC, MHT, and MFT for each month from January to December are calculated based on the 71-month OSNAP observations that cover the period from August 2014 to June 2020. For January–June and August–December, there are six monthly realizations during the 71-month observational period. For July, there are only five monthly values during the same period. The monthly climatology is calculated by averaging the available realizations of each month. The standard error of the monthly climatological mean is used as the uncertainty and is calculated as the standard deviation divided by the square root of the number of available realizations of each month (i.e., 5 for July and 6 for the other months).

**Uncertainty estimates for the time-mean MOC, MHT, and MFT.** The standard error of the 71-month mean MOC, MHT, and MFT is used as the uncertainty of the time-mean values. The standard error is calculated as:

$$SE = \frac{STD}{\sqrt{DOF}}, \quad (6)$$

where $STD$ represents the standard deviation of a quantity (i.e., the MOC, MHT, and MFT) based on the corresponding monthly time series. $DOF$ is the effective number of degrees of freedom, which is calculated as the total length of a time series divided by the decorrelation timescale. The decorrelation timescale is determined as the integral of the corresponding autocorrelation function between the first positive and negative zero-crossings[53]. We estimate that there are 43, 30, and 13 independent values in the 71-month MOC time series for the full array, OSNAP East, and OSNAP West, respectively.

**MOC calculation updates.** Several updates have been implemented in the MOC calculations since Li et al.[9] with respect to the configuration of the OSNAP array (i.e., adding or removing moorings), averaging window of the time series, and recalibration of instruments. These updates impact the calculation of the cross-sectional velocity field, which marginally changes the MOC that is calculated based on the cross-sectional velocity (Eq. 1). Here, we document each change and evaluate the impact of this change on the MOC by comparing the MOC estimates with and without this specific change.

*Averaging window.* In the previous OSNAP calculations, the MOC is calculated using 30-day averaged mooring and auxiliary data (e.g., Argo temperature and salinity profiles, wind, sea surface height, etc.)[8,9], resulting in a MOC time series with 30-day temporal resolution. As the OSNAP observations have been extended to ~6 years and will continue to a longer period, the 30-day averaging window will, in time, lead to a growing misalignment between the 30-day time series and the calendar months. Therefore, from this study onward, the MOC is calculated using monthly averaged mooring and auxiliary data to obtain an OSNAP time series aligned with calendar months, currently from August 2014 to June 2020. Since the MOC time series is calculated based on either monthly averaged or 30-day-averaged mooring and auxiliary data, a negligible difference of 0.02 Sv is detected in the 6-year mean for the full OSNAP array. However, when the monthly and 30-day MOC time series are derived from the same daily MOC time series, they have an identical time-mean value.

*Ekman transport calculation.* In the previous OSNAP metrics calculation[9], the Ekman transport across the full OSNAP section is calculated using wind stress derived from European Centre for Medium-Range Weather Forecasts (ECMWF) ERA5[54] wind speed using a bulk formula[55]. Since ERA5 directly provides wind

stress data, the Ekman transport is directly calculated using the ERA5 wind stress from this study onward. Compared to the previous calculation, using the ERA5 wind stress decreased mean MOC across the full array by only 0.06 Sv, which is much smaller than the uncertainty range.

*Inclusion of the data from moorings LSA and LSB on the West Greenland shelf.* Two additional moorings, LSA and LSB, were deployed in September of 2018 at the eastern end of the OSNAP West line to measure the inshore West Greenland Coastal Current and its water properties. The West Greenland Coastal Current carries the freshest coastal water northward into the Labrador Sea along the eastern coast. The two moorings continuously measure current velocity, temperature, and salinity throughout the water column of ~120 m. Including the two moorings in the calculation results in an increase of the full MOC by ~0.2 Sv and a decrease of southward MFT by ~0.01 Sv.

*Removal of M5 mooring in the central Irminger Sea.* M5 was a deep mooring located near the center of the Irminger Sea along the OSNAP East line. It was deployed to cover the water column from 1500 m to the sea floor from August 2014 to June 2018. In the mean, M5 captured a northward velocity over its deployment period. To assess the impact of M5's removal, a comparison between the MOC estimates with and without the data from M5 for the deployment period was performed. The result indicates that removing M5 in the calculation leads to a strengthening of the mean MOC by 0.4 Sv. Our interpretation is that M5 measured a northward recirculation branch in the MOC lower limb, which reduced the MOC estimate. For consistency with the first 4-year OSNAP calculation, we use the data from M5 in the MOC calculation from August 2014 to June 2018. After June 2018, no M5 data are available. Geostrophic velocity determined using mooring data on both sides of the original M5 position is used to fill the vacancy of M5.

*Using directly measured velocity from moorings D1, D2, and D3 along the eastern flank of the Reykjanes Ridge.* Moorings D1, D2, and D3 are located on the eastern flank of the Reykjanes Ridge in the Iceland Basin. Prior to 2018, they were configured for velocity and property measurements below 1200 m only. Velocity above 1200 m over the three moorings was calculated from geopotential height data provided by two tall moorings, one to the west and the other to the east[56]. Starting in July 2018, the three moorings were extended to cover the entire water column, allowing for the use of direct velocity measurements in the MOC calculation. A comparison of the MOC estimates for July 2018–June 2020 using geostrophic velocities with those using direct velocities shows that the latter results in a larger total MOC of just ~0.1 Sv. The MOC variability is similar for the two cases. This good agreement validates the application of end-point geostrophy over this region for the MOC estimate over the 2014–2018 period.

*IB5 mooring replaced glider measurement over the Hatton-Rockall Basin.* An additional mooring, IB5, was newly installed on the Hatton Bank along the OSNAP East section in July 2018. This mooring was added as a replacement for the discontinued glider observations with the last measurement in the Hatton-Rockall Basin in December 2017. IB5 is designed to capture the branch of the NAC in the Hatton-Rockall Basin with velocity and property measurements. A comparison between the MOC calculation with and without the IB5 data shows that its inclusion increases the mean MOC by ~0.5 Sv for the period from July 2018 to June 2020. A comparison of the MOC estimates with and without the glider data during the 2014–2017 period shows that including the glider data also increases the MOC by ~0.5 Sv. In light of the agreement between these two comparisons, we conclude that IB5 has adequately substituted for the glider measurements in the OSNAP observing system.

*Recalibration of the 53°N array along the Labrador Sea western boundary and moorings M1, M2, and M3 along the Irminger Sea western boundary.* A bias in the salinity data from the 53°N array at OSNAP West and in the Deep Western boundary moorings M1, M2, and M3 at OSNAP East for the period of August 2016–June 2018 has been attributed to a calibration error of the salinometer on board the cruise MSM74. MSM74 was responsible for the recovery and deployment of the moorings mentioned above in May and June 2018. To correct the bias in the mooring data, all affected salinities were increased by 0.004. This recalibration of the salinity data has only a minimal impact on the MOC. For both OSNAP East and OSNAP West, the difference in the mean MOC for the affected period is less than 0.1 Sv, much smaller than the uncertainties in the means.

*Labrador Shelf measurements.* As previously reported, the Labrador shelf is an unmeasured component in the OSNAP calculation[9] with velocity and property fields estimated based on monthly climatology from the multi-model ensemble and World Ocean Atlas 2018, respectively. The long-term (1950–2016) mean freshwater transport on the Labrador shelf is estimated to be about 0.06 Sv[57], accounting for more than 30% of the OSNAP West freshwater transport. To understand the impact of freshwater transport in this region on the total subpolar freshwater transport variability, two additional moorings measuring the Labrador coastal and shelf-break currents have been deployed since the summer of 2020, and four new moorings are planned to be deployed in 2023 within 100 km from the Labrador coast.

## Data availability

The 2014–2020 OSNAP MOC, MHT, and MFT time series and derived velocity, temperature, and salinity fields are available in SMARTech Repository (https://doi.org/10.35090/gatech/70342) and are freely available at www.o-snap.org/data-access/. The calibration and quality control of the moored instrument and glider data are performed by each participating group and are available at www.o-snap.org. The Argo climatology is available at http://sio-argo.ucsd.edu/RG_Climatology.html. The ECMWF ERA5 wind data are available at https://cds.climate.copernicus.eu/. The Met Office EN4.2.2 data are available at https://www.metoffice.gov.uk/hadobs/en4/. The NCEP surface heat and freshwater fluxes are available at https://rda.ucar.edu/datasets/ds094.0/.

## Code availability

The calculation code for OSNAP metrics is built based on MATLAB 2020a and is available in SMARTech Repository (https://doi.org/10.35090/gatech/70342). Detailed information about the code is available upon request to Y.F.

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

## Acknowledgements

OSNAP data were collected and made freely available by the OSNAP (Overturning in the Subpolar North Atlantic Program) project and all the national programs that contribute to it (www.o-snap.org). The moorings are part of the global OceanSITES network (http://www.oceansites.org/). Argo data were collected and made freely available by the International Argo Program and the national programs that contribute to it (http://www.argo.ucsd.edu, http://argo.jcommops.org). The Argo Program is part of the Global Ocean Observing System (https://doi.org/10.17882/42182). EN.4.2.2 data were obtained from https://www.metoffice.gov.uk/hadobs/en4/ and are © British Crown Copyright, Met Office, [2023], provided under a Non-Commercial Government Licence. We acknowledge funding from the Physical Oceanography Program of the U.S. National Science Foundation (OCE-1259398, OCE-1756231, OCE-1948335, OCE-1948482, OCE-1259618, OCE-1756363, OCE-1756361); the U.K. Natural Environment Research Council (NERC) National Capability programs CLASS (NE/R015953/1), and NERC grants UK-OSNAP (NE/K010875/1 and NE/K010875/2) and U.K. OSNAP Decade (NE/T00858X/1). Additional support was received from the European Union 7th Framework Program (FP7 2007–2013) under grant 308299 (NACLIM), the Horizon 2020 research and innovation program under grants 727852 (Blue-Action), 862626 (EuroSea). We also acknowledge support from the Royal Netherlands Institute for Sea Research and NWO grant 016.Vidi.189.130, the Surface Water and Ocean Topography-Canada (SWOT-C), Canadian Space Agency, the Aquatic Climate Change Adaptation Services Program (ACCASP), Fisheries and Oceans Canada, the Ocean Frontier Institute (OFI) and Natural Sciences and Engineering Research Council of Canada (NSERC) Discovery Grant. Support for the 53°N array by the RACE program of the German Ministry BMBF is acknowledged, as is the contribution from Fisheries and Oceans Canada's Atlantic Zone Monitoring Program. We would like to sincerely thank all the ship support offices, scientific teams, and crew members who recovered the mooring data during the COVID-19 pandemic, which made this work possible.

## Author contributions

M.S.L., A.S.B., S.A.C., B.d.Y., G.H., N.P.H., M.E.I., W.E.J., J.K., R.S.P., and F.S. conceptualized and initialized the OSNAP project. M.S.L., A.S.B., S.A.C., M.F.d.J., B.d.Y., G.H., N.P.H., M.E.I., W.E.J., J.K., R.S.P., and F.S. acquired financial support of the projects leading to this publication. M.S.L. managed and coordinated responsibilities for the research activity, planning, and execution. Y.F., T.C.B., S.A.C., F.C., M.F.d.J., B.d.Y., L.D., N.Fra., N.Fri., H.H.F., G.H., P.H., N.P.H., J.H., M.E.I., W.E.J., S.J., J.K., A.P., R.S.P., D.R., F.S., I.Y. were responsible for data collection, processing, and quality control. Y.F., F.L., M.S.L., A.S.B., S.A.C., N.P.H., and W.E.J. developed data analysis methodology. Y.F. synthesized the data and carried out data analyses. F.L. assisted in data analysis. Y.F. prepared and created the visualization/data presentation. Y.F. and M.S.L. wrote the initial draft. All authors reviewed and edited the paper.

## Competing interests

The authors declare no competing interests.
