## [Peer Review File · Communications Earth & Environment]

22nd Feb 23

Dear Dr Fu,

Please allow me once again to apologise for the delay in sending a decision on your manuscript titled "Seasonality of the Meridional Overturning Circulation in the Subpolar North Atlantic", which was due to the need to replace a reviewer at a late stage. The paper has now been seen by two reviewers, and I include their comments at the end of this message. They find your work of interest, but some important points are raised. We are interested in the possibility of publishing your study in *Communications Earth & Environment*, but would like to consider your responses to these concerns and assess a revised manuscript before we make a final decision on publication.

We therefore invite you to revise and resubmit your manuscript, along with a point-by-point response that takes into account the points raised. In order for the paper to be suitable for *Communications Earth & Environment*, we will need you to:

- * Justify the robustness of your conclusions on seasonality, in light of a short record (6 years) that is dominated by a NAO+ phase.
- * Discuss the results and underlying mechanisms more thoroughly in the context of the existing literature (including earlier RAPID studies and modelling studies) and clearly bring out the novelty of the study.

We recommend that you remove all claims of "first": these can lead to unproductive controversy and are unnecessary. Please just state your results in comparison with the existing literature.

Please highlight all changes in the manuscript text file (or send a list of changes in the manuscript text in response to the reports).

Please use the following link to submit your revised manuscript, point-by-point response to the referees' comments (which should be in a separate document to any cover letter) and the completed checklist:

[link redacted]

We hope to receive your revised paper within six weeks; please let us know if you aren't able to submit it within this time so that we can discuss how best to proceed. If we don't hear from you, and the revision process takes significantly longer, we may close your file. In this event, we will still be happy to reconsider your paper at a later date, as long as nothing similar has been accepted for publication at Communications Earth & Environment or published elsewhere in the meantime.

We understand that due to the current global situation, the time required for revision may be longer than usual. We would appreciate it if you could keep us informed about an estimated timescale for resubmission, to facilitate our planning. Of course, if you are unable to estimate, we are happy to accommodate necessary extensions nevertheless.

Please do not hesitate to contact me if you have any questions or would like to discuss these revisions further. We look forward to seeing the revised manuscript and thank you for the opportunity to review your work.

Best regards,

Heike Langenberg, PhD

Chief Editor

Communications Earth & Environment

On Twitter: @CommsEarth

EDITORIAL POLICIES AND FORMATTING

Editorial Policy: [Policy requirements](https://www.nature.com/documents/nr-editorial-policy-checklist.pdf) (Download the link to your computer as a PDF.)

Furthermore, please align your manuscript with our format requirements, which are summarized on the following checklist:

[Communications Earth & Environment formatting checklist](https://www.nature.com/documents/commsj-phys-style-formatting-checklist-article.pdf)

and also in our style and formatting guide [Communications Earth & Environment formatting guide](https://www.nature.com/documents/commsj-phys-style-formatting-guide-accept.pdf) .

*** DATA: Communications Earth & Environment endorses the principles of the Enabling FAIR data project (<http://www.copdess.org/enabling-fair-data-project/>). We ask authors to make the data that support their conclusions available in permanent, publically accessible data repositories. (Please contact the editor if you are unable to make your data available).

All Communications Earth & Environment manuscripts must include a section titled "Data Availability" at the end of the Methods section or main text (if no Methods). More information on this policy, is available at <http://www.nature.com/authors/policies/data/data-availability-statements-data-citations.pdf>.

DATA SOURCES: All new data associated with the paper should be placed in a persistent repository where they can be freely and enduringly accessed. We recommend submitting the data to discipline-specific, community-recognized repositories, where possible and a list of recommended repositories is provided at a

<http://www.nature.com/sdata/policies/repositories>><http://www.nature.com/sdata/policies/repositories>.

If a community resource is unavailable, data can be submitted to generalist repositories such as [figshare](https://figshare.com/) or [Dryad Digital Repository](http://datadryad.org/). Please provide a unique identifier for the data (for example a DOI or a permanent URL) in the data availability statement, if possible. If the repository does not provide identifiers, we encourage authors to supply the search terms that will return the data. For data that have been obtained from publically available sources, please provide a URL and the specific data product name in the data availability statement. Data with a DOI should be further cited in the methods reference section.

Please refer to our data policies at <http://www.nature.com/authors/policies/availability.html>><http://www.nature.com/authors/policies/availability.html>.

REVIEWER COMMENTS:

Reviewer #1 (Remarks to the Author):

The manuscript describes the seasonality of the local MOC, MHT, and MFT under the OSNAP array. It should be categorized as one contribution of the many OSNAP papers in recent years. The insights of the paper will influence the scientific community because the seasonality under OSNAP is discussed the first time. The manuscript is well written and the described methods are sound. I suggest that the present manuscript should be subject to minor revision. I believe the contextual knowledge should be enhanced at some lines of the manuscript. Below I provide minor comments.

A weakness of the study is that the interrelations between MOC, MHT, and MFT are not well explained. The interrelation is mainly described in one direction in the sense that the influence of the local MOC on MHT and MFT is discussed. For instance, I believe there is a lack in description of summertime atmospheric conditions or MFT on local ocean dynamics. Moreover, the summer states are not well explained.

As a matter of concern, the authors should state explicitly whether they believe that the seasonality found in this study also describes near-term future states in light of climate change.

Another question that arises is whether there is a direct relation between the seasonality of the local MOC and the mixed-layer depths of the convection regions.

A question that is difficult to answer: I would find it interesting to have a statement on the upstream influence of the upper MOC on the local MOC, MHT, and MFT. There is lack of meridional coherence of MOC variability on this timescale, but it would be worth to note whether near-sided locations do influence the local MOC, MFT, and MHT seasonality.

Please discuss major differences the reference 12 and 13, as these studies are closely related to your study.

As from line 70, it should be mentioned which methods are employed to e.g. compute the geostrophic velocities under OSNAP. I believe it is worth mentioning Baehr (2004).

I believe Fig. 4b is already discussed in previous studies. Maybe it would be better to describe the insights of Fig. 4b by words only?

As from line 387, the summertime influence on MFT is not discussed, which makes this section on MFT inconsistent.

Reviewer #3 (Remarks to the Author):

Overall, positive remarks for the manuscript, but I have some questions regarding the statistics, how these findings relate to the current understanding of AMOC in the modelling community, and the novelty of the results. AMOC has been the focus to many recent studies, both with observations and models. I have trouble being convinced that from 6 years' worth of data, there is enough to know the seasonal variability. Particularly with the lack of comparison to modeled variability, RAPID/MOCHA, and the fact all the observations take place during a specific NAO phase. I have left several remarks below for the authors to consider. I think at the current state, the manuscript reads well and presents an important storyline, but should expand on their discussion and references to complete their story and better define how these results are novel compared to previous AMOC studies (both obs. and model) before being considered for publication. At the current stage, I would suggest manuscript revisions.

1. The manuscript focuses on being the first to demonstrate the overturning variability. First, I think more emphasis should be put on stating overturning variability from the subpolar North Atlantic. For example, even in the abstract, the text states “first observational evidence of overturning variability on seasonal and longer timescales”. Although I would agree this is true for the subpolar region, Moat et al., 2020 (<https://os.copernicus.org/articles/16/863/2020/#section4>) has done a very good job demonstrating this pattern in the subtropical Atlantic. Please be specific throughout the text in making key statements regarding the exact region. I think differentiating this subpolar AMOC from the subtropical AMOC (i.e., RAPID / MOCHA) is crucial. Moreover, RAPID / MOCHA is mentioned in the Introduction, but I think it could be very useful to compare the magnitude of seasonal variability.

2. In lines 61-64 there is reference to modeling studies linking the redistribution of energy with AMOC states and dense water mass formation in the subpolar North Atlantic, but the discussion from the modelling community is minimal after these lines. For example, Menary et al (2020; <https://doi.org/10.1029/2020GL089793>) discusses the role of the Labrador Sea in AMOC variability from OSNAP and in models, but there is no mention of this study (Perhaps close to line 80 or 227?). Moreover, I think there should at least be mention to Oldenburg et al (2020; 2021; <https://doi.org/10.1175/JCLI-D-20-0614.1>; <https://doi.org/10.1029/2021JC018102>) which discusses MHT and water mass transformation within the North Atlantic subpolar region, using both models and observations.

3. From line 128, the determination is now able to be made from 6 years? Is there any statistical backing to suggest 6 years is enough compared to 4 years to identify the seasonal variability with a strong confidence? From line 96, the previous timeseries was too short, but in line 98, it is said the signal is clearly emerging. To me, I don't read that as a statement of strong confidence to later state that it is possible to identify the seasonal variability. I think the wording should reflect the degree of confidence (i.e. using the word “suspect” in line 423).

4. In line 157+, it is stated that the OSNAP West actually recorded the largest amount of overturning contribution during the last two years. Is there any anomalous sea ice state in the Labrador / Irminger sea regions that could be related to the OSNAP West state?

5. Line 167, how is the seasonal cycle statistically significant? Significant based on the standard error? I think the authors should point to the section of the methods.

6. Figure 2, I think there should be greater detail to the Monte Carlo analysis rather than just putting a reference. Is there replacement? Accounting for any degree of autocorrelation?

7. Line 170, in describing the maximum overturning, is there any relationship to peak MOC to max / min sea Arctic ice state?

8. With the general use of using coupled climate models to help understand and predict potential climate states resulting from Anthropogenic forcing, do you find similar OSNAP West and East drivers or mean states to what is summarized in Heuzé et al (2017, 2021; <https://os.copernicus.org/articles/13/609/2017/os-13-609-2017-discussion.html>; <https://os.copernicus.org/articles/17/59/2021/>) ? Do the climate models generally experience deep convection or dense water formation like your results?

9. From Ortega et al 2015, they explore North Atlantic subpolar variability from Ekman transports, compare the overturning index (AMOC minus Ekman), and compare with winter mixed layer variability. There are several instances of stating to be the first, but several of the analyses proposed have been done in the modeling community. I think the authors should rephrase to state being the first with observed subpolar overturning or go back and compare with previous modeling studies. Ortega et al 2015 is just an example (https://www.sciencedirect.com/science/article/pii/S0079661115001482?casa_token=t6pc1-3BJRAAAAAA:NI-1pgjnBaLc-CLWsBhgNQIUHmMnEBjARhGNJbLMJD_SdDNL1O6Xc7PCzcuDouoQm3QLZYG2szlJ) . But for example, in line 216, there is a peak transformation between December to March. This strongly agrees with the use in models that the deep convective regions can be identified from winter MLD variability. Perhaps no reference need, but it is good to know that climate modelers have been using a technique that matches the observed data from the subpolar North Atlantic.

10. Within the summary section, it is brought up that the 2014-2020 period was mostly dominated by a winter NAO+ positive phase, with strong links to the AMOC state (density and wind driven). Personally, I think this should have been brought up much earlier in the text (perhaps after line 145). After, it is discussed that the most recent two years have been an NAO- phase, which the data has not yet been retrieved. Knowing this, how do you think the results will change? From modelling, changing the NAO phase can have a large impact on the AMOC and subpolar region (i.e., Ferster et al., 2022 induced NAO state and sea ice decline; <https://doi.org/10.1029/2022GL097967>). Do you think this change in NAO phase could play a large role in the observed variability? Are 6 years (even 8) enough to say with confidence to know the observed seasonal variability, being that there is low-frequency anthropogenic forcing and in a region of strong decadal-to-multidecadal variability? I think the summary should better clarify and explain the potential role of NAO on the variability.

11. Line 436: I think this should be interannual to decadal, to even multidecadal.

12. At the end of the manuscript, the authors state this study will be important to ground truthing for models, but there is minimal comparison to models throughout the manuscript. Do models do well for a mean state? Variability? Although I agree this study will be important for the modeling community, but then there at least should be some reference to the current quality / state of models.

13. Other potential useful references on modeled variability in the subpolar North Atlantic: Mielke et al 2013 (<https://doi.org/10.1002/grl.50233>) and Xu et al 2014 (<https://doi.org/10.1002/2014JC009994>)

**Authors' response to reviewer comments on the manuscript "Seasonality of the Meridional**
**Overturning Circulation in the Subpolar North Atlantic"**

Yao Fu, M. Susan Lozier, Tiago Carrilho Biló, Amy S. Bower, Stuart A. Cunningham, Frédéric Cyr,
5 M. Femke de Jong, Brad DeYoung, Lewis Drysdale, Neil Fraser, Nora Fried, Heather H. Furey,
Guoqi Han, Patricia Handmann, N. Penny Holliday, James Holte, Mark E. Inall, William E. Johns,
Sam Jones, Johannes Karstensen, Feili Li, Astrid Pacini, Robert S. Pickart, Darren Rayner,
Fiammetta Straneo, Igor Yashayaev

We would like to thank both reviewers for their thorough examination of the manuscript. The
comments are constructive and have helped us considerably improve the manuscript.

Below are our responses to the reviews. Each reviewer's comment (in black font) is followed by
our response (in blue font). The corresponding changes to the manuscript are indicated after
each response. Note that the referenced line numbers below correspond in all cases to those in
the revised article file.

REVIEWER COMMENTS:

Reviewer #1 (Remarks to the Author):

The manuscript describes the seasonality of the local MOC, MHT, and MFT under the OSNAP array.
It should be categorized as one contribution of the many OSNAP papers in recent years. The
insights of the paper will influence the scientific community because the seasonality under
OSNAP is discussed the first time. The manuscript is well written and the described methods are
sound. I suggest that the present manuscript should be subject to minor revision. I believe the
contextual knowledge should be enhanced at some lines of the manuscript. Below I provide
minor comments.

Thank you for the positive and constructive comments. We have revised the manuscript to
account for your suggestions. Please see our detailed responses below.

A weakness of the study is that the interrelations between MOC, MHT, and MFT are not well
explained. The interrelation is mainly described in one direction in the sense that the influence
of the local MOC on MHT and MFT is discussed. For instance, I believe there is a lack in description
of summertime atmospheric conditions or MFT on local ocean dynamics. Moreover, the summer
states are not well explained.

We have enhanced the discussion of the summer MFT state as well as the impact of summertime
atmosphere conditions on ocean dynamics. Please see lines 388-398 and lines 404-407.

Please also refer to our response to your last comment for more detail (lines 119-127 in this
response document).

As a matter of concern, the authors should state explicitly whether they believe that the
seasonality found in this study also describes near-term future states in light of climate change.

In the summary, we now point out that the seasonal cycle is derived from the 6-year OSNAP
observations and that as the atmospheric conditions change in the coming years and decades, it
is likely that the strength of the surface transformation and Ekman transport will also change. As
a result, MOC, MHT, and MFT seasonal variability may also change. However, given the inherent
seasonality of atmospheric forcing, i.e., the westerly wind and surface buoyancy forcing, the
timing of the seasonal cycles for MOC, MHT, and MFT should remain the same. Please see lines
427-439.

Another question that arises is whether there is a direct relation between the seasonality of the
local MOC and the mixed-layer depths of the convection regions.

We think that the seasonality of mixed layer depth (MLD) and the seasonality of the local
overturning in a convective basin (e.g., the Labrador Sea) are both driven by the seasonality of
atmospheric forcing. Holte and Straneo (2017) show that the MLD reaches a maximum in March
due to continuous winter cooling and decreases to a minimum in July due to restratification in
spring and early summer. This pattern is consistent with the seasonal cycle of water mass
transformation, which shows continuous wintertime formation of dense water until March.
However, the MLD describes the thickness of the mixed layer, while transformation describes the
process that converts lighter water to denser water due to buoyancy forcing. Given that the
overturning circulation is a measure of the volume conversion from lighter to denser waters, we
focus on transformation instead of MLD in this study as it has a direct link with the MOC.

A question that is difficult to answer: I would find it interesting to have a statement on the
upstream influence of the upper MOC on the local MOC, MHT, and MFT. There is lack of
meridional coherence of MOC variability on this timescale, but it would be worth to note whether
near-sided locations do influence the local MOC, MFT, and MHT seasonality.

Other than the impact on MFT because of the seasonality in the freshwater along the Greenland
and Labrador shelves/slopes, which we note in the paper, the impact on MOC, MFT and MHT
from upstream or 'non-local' sources is most likely on interannual to decadal time scales. To that
point, some recent studies have explored non-local impacts on the recently observed freshening
in the subpolar North Atlantic (Holliday et al. 2020). From a study with a high-resolution model,
Fox et al. (2022) suggest that the freshening in the broader subpolar region is triggered by
reduced heat loss in the Labrador Sea about a decade ago. Bilo et al. (2022) use OSNAP
observations to further show that the freshening signal has arrived in the Irminger basin and
weakened local convection there. However, the exact impact on overturning is under
investigation. Please see lines 464-468.

Furthermore, based on your comment, we now compare the seasonal cycles of the MOC in the
subpolar and subtropical regions. In the subtropics, wind dynamics dominate the MOC seasonal

cycle, which has its maximum in autumn and minimum in spring. As the reviewer noted, the
subtropical MOC influence on the subpolar MOC is unlikely on seasonal time scales. Please see
the comparison in lines 421-424.

Please discuss major differences the reference 12 and 13, as these studies are closely related to
your study.

The focus of our study is on MOC seasonality. Therefore, to avoid any confusion we have decided
to remove the citation to Feucher et al. (2019), which primarily investigates the relationship
between the Labrador Sea water formation rate and Labrador Sea overturning strength on time
scales longer than seasonal.

As from line 70, it should be mentioned which methods are employed to e.g. compute the
geostrophic velocities under OSNAP. I believe it is worth mentioning Baehr (2004).

We now mention in Method that geostrophic transports in the basin interior are calculated from
dynamic height based on the thermal wind relation and cite Baehr (2004). Please see lines 490-
492.

I believe Fig. 4b is already discussed in previous studies. Maybe it would be better to describe the
insights of Fig. 4b by words only?

We appreciate this suggestion but believe that Fig. 4b provides readers a useful geographic
reference and information about the updated mooring locations and depths. In addition, the
mean salinity section in Fig. 4b together with Fig. 4a will help readers identify which regions are
expected to have a higher contribution to MFT variability. Therefore, we prefer to keep Fig. 4b in
the paper.

As from line 387, the summertime influence on MFT is not discussed, which makes this section
on MFT inconsistent.

We have revised the paragraph to better describe the seasonal cycle of the overturning and gyre
contribution to MFT at OSNAP East. Please see lines 388-398 and lines 404-406.

As we explain in those added lines, the overturning and gyre contributions to MFT are both at
their minimum at OSNAP East during the summer due to the fact that warm temperatures and
land/sea ice melting stratify the water column, leading to weaker local overturning. Weak winds
in summer also decrease the gyre circulation strength. In addition, we also emphasize that at
OSNAP West, the summer gyre MFT contribution is the strongest due to fresh coastal water
export along the western boundary, which is likely a result of summer ice melting.

Reviewer #3 (Remarks to the Author):

Overall, positive remarks for the manuscript, but I have some questions regarding the statistics, how these findings relate to the current understanding of AMOC in the modelling community, and the novelty of the results. AMOC has been the focus to many recent studies, both with observations and models. I have trouble being convinced that from 6 years' worth of data, there is enough to know the seasonal variability. Particularly with the lack of comparison to modeled variability, RAPID/MOCHA, and the fact all the observations take place during a specific NAO phase. I have left several remarks below for the authors to consider. I think at the current state, the manuscript reads well and presents an important storyline, but should expand on their discussion and references to complete their story and better define how these results are novel compared to previous AMOC studies (both obs. and model) before being considered for publication. At the current stage, I would suggest manuscript revisions.

Thank you for the positive and constructive review. Please see our detailed responses below.

1. The manuscript focuses on being the first to demonstrate the overturning variability. First, I think more emphasis should be put on stating overturning variability from the subpolar North Atlantic. For example, even in the abstract, the text states “first observational evidence of overturning variability on seasonal and longer timescales”. Although I would agree this is true for the subpolar region, Moat et al., 2020 (<https://os.copernicus.org/articles/16/863/2020/#section4>) has done a very good job demonstrating this pattern in the subtropical Atlantic. Please be specific throughout the text in making key statements regarding the exact region. I think differentiating this subpolar AMOC from the subtropical AMOC (i.e., RAPID / MOCHA) is crucial. Moreover, RAPID / MOCHA is mentioned in the Introduction, but I think it could be very useful to compare the magnitude of seasonal variability.

We now specify that this study presents the first observational evidence of MOC seasonality in the subpolar North Atlantic and compare the subpolar MOC seasonal cycle with that of the subtropical MOC. Please see lines 56-58, 114-116 and 421-424.

2. In lines 61-64 there is reference to modeling studies linking the redistribution of energy with AMOC states and dense water mass formation in the subpolar North Atlantic, but the discussion from the modelling community is minimal after these lines. For example, Menary et al (2020; <https://doi.org/10.1029/2020GL089793>) discusses the role of the Labrador Sea in AMOC variability from OSNAP and in models, but there is no mention of this study (Perhaps close to line 80 or 227?). Moreover, I think there should at least be mention to Oldenburg et al (2020; 2021; <https://doi.org/10.1175/JCLI-D-20-0614.1>; <https://doi.org/10.1029/2021JC018102>) which discusses MHT and water mass transformation within the North Atlantic subpolar region, using both models and observations.

We have added discussions of the modeling studies suggested by the reviewer. Please see lines
150-152, 436-437, and 473-475.

3. From line 128, the determination is now able to be made from 6 years? Is there any statistical
backing to suggest 6 years is enough compared to 4 years to identify the seasonal variability with
a strong confidence? From line 96, the previous timeseries was too short, but in line 98, it is said
the signal is clearly emerging. To me, I don't read that as a statement of strong confidence to later
state that it is possible to identify the seasonal variability. I think the wording should reflect the
degree of confidence (i.e. using the word "suspect" in line 423).

To address your concern, we calculated the signal-to-noise ratio (SN) of the seasonal cycle. We
define SN as the difference between the maximum and minimum divided by the mean of the
uncertainties of the maximum and minimum.

Using the 4-year data, the SNs for the MOC seasonal cycle across the full OSNAP section, OSNAP
East, and OSNAP West are 6.7, 6.2, and 2.9, respectively. Using the 6-year data, the SNs are 7.4,
6.8, and 4.0, respectively. We understand that these differences, though favorable to the 6-year
time series, are not drastically different. Thus, we have reworded the text as:

*"Using the 71-month (~6 years) OSNAP record now in hand, the overall goal of this study is to*
*ascertain the seasonal cycle of the subpolar MOC, MHT and MFT in order to quantify how much*
*of this large month-to-month variability can be attributed to seasonality."*

In short, we no longer insinuate or make the claim that 6 years is much more favorable for a
seasonal assessment than 4 years. We instead focus on the statistical significance of the 6-year
seasonality in and of itself. Thanks for raising this concern as it caused us to quantify the
difference and be more careful with our language.

4. In line 157+, it is stated that the OSNAP West actually recorded the largest amount of
overturning contribution during the last two years. Is there any anomalous sea ice state in the
Labrador / Irminger sea regions that could be related to the OSNAP West state?

We are not aware of an anomalous sea ice state in the last two years (2019-2020) of the OSNAP
record. Yashayaev et al. (2021) produce a time series of ice extent anomaly from 1978 to 2018 in
the northern Labrador Sea, which shows that the ice extent was clearly smaller in the recent
decade compared to the 1980s and to the first half of the 1990s, during which the NAO was
strongly positive. With our next two years of data, our intended focus is on the driving
mechanisms governing interannual variability. With that analysis, we will have the opportunity
to more fully explore any relationship between winds, buoyancy forcing, sea ice state, etc.

For this paper focused on seasonality, we are not prepared to attribute the recent OSNAP West
increase to any one mechanism.

5. Line 167, how is the seasonal cycle statistically significant? Significant based on the standard error? I think the authors should point to the section of the methods.

We have modified the text to now specify that the MOC maximum is significantly larger than the MOC minimum by showing the values \pm uncertainty. We also point to the Methods section for standard error estimation. Here is the modified text from lines 167-171:

“A seasonal cycle of the SPNA MOC is extracted by calculating the climatological mean MOC for each month using the 6-year OSNAP time series (Fig. 3a, see Methods). Across the full OSNAP section, the seasonal MOC maximum in spring (21.1 ± 1.2 Sv) is significantly larger than the minimum in winter (12.3 ± 1.2 Sv) with a peak-to-trough change of 8.8 Sv (Table S2, see Methods for the standard error estimation).”

6. Figure 2, I think there should be greater detail to the Monte Carlo analysis rather than just putting a reference. Is there replacement? Accounting for any degree of autocorrelation?

For the Monte Carlo analysis, we assume that all variables used for the calculation of MOC, MHT and MFT follow a normal distribution. For each monthly estimate of MOC, MHT, and MFT, we create a normally-distributed field of variables (temperature, salinity, velocity, wind stress, etc.) using the monthly mean and standard deviation. For each Monte Carlo iteration, we randomly draw from the distribution and produce one realization of MOC, MHT and MFT. We run the Monte Carlo simulation until the running means of the MOC, MHT, and MFT estimates converge to a value smaller than a predefined value. We then use the mean and standard deviation of all the realizations as the monthly mean and uncertainty of the MOC, MHT and MFT.

This Monte Carlo analysis is consistent with two previous OSNAP publications (Lozier et al., 2019 and Li et al., 2021) and no change has been applied to this method. Therefore, in this study we do not provide details of the Monte Carlo analysis, rather we refer the reader to Lozier et al. (2019) for an explanation of the Monte Carlo method that we employ.

7. Line 170, in describing the maximum overturning, is there any relationship to peak MOC to max / min sea Arctic ice state?

On seasonal time scales, Arctic sea ice extent and the subpolar MOC are driven by the annual cycle of atmospheric forcing, but the two quantities are likely not directly related due to the distance and time required for anomalies to propagate from one location to the other. On longer time scales, direct observations are lacking. Model simulations suggest that the relationship between Arctic sea ice and the MOC is complex. For instance, Delworth et al. (2016) shows that a stronger MOC increases the northward oceanic heat transport, thus reducing Arctic sea ice extent. Similarly, Liu et al. (2020) found that a weakened MOC will slow Arctic sea ice loss in the future also because of reduced poleward heat transport. On the other hand, sea ice loss increases freshwater in the subpolar basins and enhances absorbed solar radiation, which negatively impacts MOC strength.

Because the focus of our paper is on seasonality, we have added only a brief mention of the complicated relationship between sea ice and the MOC that has been gleaned from modeling studies. Continuing OSNAP observations may give answers to some of the questions from an observational perspective. Please see lines 462-464.

8. With the general use of using coupled climate models to help understand and predict potential climate states resulting from Anthropogenic forcing, do you find similar OSNAP West and East drivers or mean states to what is summarized in Heuzé et al (2017, 2021; <https://os.copernicus.org/articles/13/609/2017/os-13-609-2017-discussion.html>; <https://os.copernicus.org/articles/17/59/2021/>) ? Do the climate models generally experience deep convection or dense water formation like your results?

In this study, we find that transformation in both the western and eastern subpolar gyre is a key driver for the overturning circulation seasonal cycle. In comparison, CMIP5 and CMIP6 models show a spread in the locations and strength of deep water formation, according to Heuze et al. (2017, 2021). For instance, some of the CMIP6 models only have deep convection in either the Labrador Sea or Irminger Sea, some models have deep convection in the Labrador and Irminger Seas as two separate regions, while nearly a half of the models have convection over a broad area in the subpolar region (not distinguishing the western and eastern subpolar gyre). The strength of convection also differs among the models. In general, the stronger the convection, the stronger the MOC in the mean state.

We now add a brief mention in the summary about climate models' performance in simulating the subpolar MOC on lines 473-475. We also point out that a few models can reproduce the observed time-mean OSNAP East and OSNAP West overturning strength, such as the GFDL model (Zhang et al., 2021), HadGEM3-GC3.1-MM (Menary et al., 2020), and the high-resolution version of CESM (Yeager et al., 2021) on lines 150-152. In addition, the connection between Labrador Sea water production and downstream MOC has been investigated previously using model simulations (Li et al., 2019).

9. From Ortega et al 2015, they explore North Atlantic subpolar variability from Ekman transports, compare the overturning index (AMOC minus Ekman), and compare with winter mixed layer variability. There are several instances of stating to be the first, but several of the analyses proposed have been done in the modeling community. I think the authors should rephrase to state being the first with observed subpolar overturning or go back and compare with previous modeling studies. Ortega et al 2015 is just an example (https://www.sciencedirect.com/science/article/pii/S0079661115001482?casa_token=t6pc1-3BJRAAAAAA). But for example, in line 216, there is a peak transformation between December to March. This strongly agrees with the use in models that the deep convective regions can be identified from winter MLD variability. Perhaps no reference need, but it is good to know that climate modelers have been using a technique that matches the observed data from the subpolar North Atlantic.

We have rephrased several statements throughout the main text and specified that this work is
an observation-based study focusing on the subpolar region.

10. Within the summary section, it is brought up that the 2014-2020 period was mostly
dominated by a winter NAO+ positive phase, with strong links to the AMOC state (density and
wind driven). Personally, I think this should have been brought up much earlier in the text
(perhaps after line 145). After, it is discussed that the most recent two years have been an NAO-
phase, which the data has not yet been retrieved. Knowing this, how do you think the results will
change? From modelling, changing the NAO phase can have a large impact on the AMOC and
subpolar region (i.e., Ferster et al., 2022 induced NAO state and sea ice
decline; <https://doi.org/10.1029/2022GL097967>). Do you think this change in NAO phase could
play a large role in the observed variability? Are 6 years (even 8) enough to say with confidence
to know the observed seasonal variability, being that there is low-frequency anthropogenic
forcing and in a region of strong decadal-to-multidecadal variability? I think the summary should
better clarify and explain the potential role of NAO on the variability.

In response to your concern, we have enhanced our discussion of NAO's role in MOC variability
in the Summary. Please see lines 427-439. In general, positive NAO is associated with strong
westerly winds and intense surface cooling in winter. As a result, on seasonal time scales, the
magnitude of the MOC seasonal cycle (seasonal maximum to minimum) may change during
different NAO phases. However, the timing of the seasonal cycle should remain the same. On
longer time scales (i.e., interannual to decadal), based on modeling studies and observations of
wind and buoyancy forcing, we can reasonably expect that the MOC strength may change under
different NAO conditions. The 8-year timeseries should reveal certain characteristics of subpolar
MOC interannual variability.

11. Line 436: I think this should be interannual to decadal, to even multidecadal.

We changed the phrase to "interannual to multidecadal", now on line 451.

12. At the end of the manuscript, the authors state this study will be important to ground truthing
for models, but there is minimal comparison to models throughout the manuscript. Do models
do well for a mean state? Variability? Although I agree this study will be important for the
modeling community, but then there at least should be some reference to the current quality /
state of models.

We have expanded our reference to modeling studies. For example, we include a discussion about
NAO's impact on MOC variability based on model works; we mention the complex relationship
between Arctic sea ice and the MOC indicated by modeling studies; and we cite studies about
model performance in simulating the MOC.

13. Other potential useful references on modeled variability in the subpolar North Atlantic: Mielke
et al 2013 (<https://doi.org/10.1002/grl.50233>) and Xu et al 2014
(<https://doi.org/10.1002/2014JC009994>)

These two papers are now cited in the Introduction with respect to MOC seasonality in the
subtropics.

Reference:

Holte, J. & Straneo, F. Seasonal Overturning of the Labrador Sea as Observed by Argo Floats. *J.*
*Phys. Oceanogr.* **47**, 2531–2543 (2017).

Holliday, N. P. *et al.* Ocean circulation causes the largest freshening event for 120 years in
eastern subpolar North Atlantic. *Nat. Commun.* **11**, 585 (2020).

Fox, A. D. *et al.* Exceptional freshening and cooling in the eastern subpolar North Atlantic caused
by reduced Labrador Sea surface heat loss. *Ocean Sci.* **18**, 1507–1533 (2022).

Biló, T. C., Straneo, F., Holte, J. & Le Bras, I. A. A. Arrival of New Great Salinity Anomaly Weakens
Convection in the Irminger Sea. *Geophys. Res. Lett.* **49**, 1–10 (2022).

Yashayaev, I., Peterson, I., & Wang, Z. (2021). Meteorological, Sea Ice, and Physical
Oceanographic Conditions in the Labrador Sea during 2018. *DFO Can. Sci. Advis. Sec. Res.*
*Doc., 2021/042*, iv + 26 p.

Delworth, T. L. *et al.* The North Atlantic Oscillation as a driver of rapid climate change in the
Northern Hemisphere. *Nat. Geosci.* **9**, 509–512 (2016).

Liu, W., Fedorov, A. V., Xie, S. P. & Hu, S. Climate impacts of a weakened Atlantic meridional
overturning circulation in a warming climate. *Sci. Adv.* **6**, 1–9 (2020).

Lozier, M. S. *et al.* A sea change in our view of overturning in the subpolar North Atlantic.
*Science (80-.).* **363**, 516–521 (2019).

Li, F. *et al.* Subpolar North Atlantic western boundary density anomalies and the Meridional
Overturning Circulation. *Nat. Commun.* **12**, 1–9 (2021).

Li, F., M. S. Lozier, G. Danabasoglu, N. P. Holliday, Y. Kwon, A. Romanou, S. G. Yeager, and R. Zhan.
Local and Downstream Relationships between Labrador Sea Water Volume and North
Atlantic Meridional Overturning Circulation Variability. *J. Climate*, **32**, 3883–3898,

Zhang, R. & Thomas, M. Horizontal circulation across density surfaces contributes substantially
to the long-term mean northern Atlantic Meridional Overturning Circulation. *Commun.*
*Earth Environ.* **2**, (2021).

Heuzé, C. North Atlantic deep water formation and AMOC in CMIP5 models. *Ocean Sci* **13**, 609–
622 (2017).

Heuzé, C. Antarctic Bottom Water and North Atlantic Deep Water in CMIP6 models. *Ocean Sci.*
**17**, 59–90 (2021).

Menary, M. B., Jackson, L. C. & Lozier, M. S. Reconciling the Relationship Between the AMOC
and Labrador Sea in OSNAP Observations and Climate Models. *Geophys. Res. Lett.* **47**,
(2020).

Yeager, S. G., Karspeck, A., Danabasoglu, G., Tribbia, J. & Teng, H. A decadal prediction case
study: Late twentieth-century north Atlantic Ocean heat content. *J. Clim.* **25**, 5173–5189
(2012).

19th Apr 23

Dear Dr Fu,

Your manuscript titled "Seasonality of the Meridional Overturning Circulation in the Subpolar North Atlantic" has now been seen by our reviewers, whose comments appear below. In light of their advice I am delighted to say that we are happy, in principle, to publish a suitably revised version in Communications Earth & Environment under the open access CC BY license (Creative Commons Attribution v4.0 International License).

We therefore invite you to revise your paper one last time to address the remaining concerns of our reviewers. At the same time we ask that you edit your manuscript to comply with our format requirements and to maximise the accessibility and therefore the impact of your work.

EDITORIAL REQUESTS:

*****Please take care to match our formatting and policy requirements. We will check revised manuscript and return manuscripts that do not comply. Such requests will lead to delays. *****

SUBMISSION INFORMATION:

OPEN ACCESS:

Communications Earth & Environment is a fully open access journal. Articles are made freely accessible on publication under a [CC BY license](http://creativecommons.org/licenses/by/4.0) (Creative Commons Attribution 4.0 International License). This license allows maximum dissemination and re-use of open access materials and is preferred by many research funding bodies.

For further information about article processing charges, open access funding, and advice and support from Nature Research, please visit <https://www.nature.com/commsenv/article-processing-charges>

At acceptance, you will be provided with instructions for completing this CC BY license on behalf of all authors. This grants us the necessary permissions to publish your paper. Additionally, you will be asked to declare that all required third party permissions have been obtained, and to provide billing information in order to pay the article-processing charge (APC).

[link redacted]

Best regards,

Heike Langenberg, PhD

Chief Editor

Communications Earth & Environment

On Twitter: @CommsEarth

REVIEWERS' COMMENTS:

Reviewer #1 (Remarks to the Author):

My minor and specific comments are now met.

Reviewer #3 (Remarks to the Author):

This is the second review of the manuscript entitled "Seasonality of the Meridional Overturning Circulation in the Subpolar North Atlantic" by Fu et al. I believe the authors have addressed the previous concerns and comments raised by the three reviewers and the results are new in the context of observed seasonality of the North Atlantic overturning. I believe the manuscript should be accepted and considered for publication.